# Cloud Droplet Growth in Shallow Cumulus Clouds Considering 1D and 3D Thermal Radiative Effects

Carolin Klinger[1,2], Graham Feingold[1], and Takanobu Yamaguchi[1,3]

[1]Chemical Sciences Division, NOAA Earth System Research Laboratory (ESRL), Boulder, Colorado, USA
[2]Ludwig-Maximilians-Universität München, Lehrstuhl für Experimentelle Meteorologie
[3]Cooperative Institute for Research in Environmental Sciences, University of Colorado, Boulder, Boulder, CO, USA

**Correspondence:** Carolin Klinger (carolin.klinger@physik.lmu.de)

**Abstract.** The effect of 1D and 3D thermal radiation on cloud droplet growth in shallow cumulus clouds is investigated using large eddy simulations with size resolved cloud microphysics. A two step approach is used for separating microphysical effects from dynamical feedbacks. In step one, an offline parcel model is used to describe the onset of rain. The growth of cloud droplets to raindrops is simulated with bin-resolved microphysics along previously recorded Lagrangian trajectories. It is shown that thermal heating and cooling rates can enhance droplet growth and raindrop production. Droplets grow to larger size bins in the 10-30 $\mu$m radius range. The main effect in terms of raindrop production arises from recirculating parcels, where a small number of droplets is exposed to strong thermal cooling at cloud edge. These recirculating parcels, comprising about 6-7% of all parcels investigated, make up 45% of the rain for the no radiation simulation and up to 60% when 3D radiative effects are considered. The effect of 3D thermal radiation on rain production is stronger than that of 1D thermal radiation. 3D thermal radiation can enhance the rain amount up to 40% compared to standard droplet growth without radiative effects in this idealized framework.

In the second stage, fully coupled large eddy simulations show that dynamical effects are stronger than microphysical effects, as far as the production of rain is concerned. 3D thermal radiative effects again exceed 1D thermal radiative effects. Small amounts of rain are produced in more clouds (over a larger area of the domain) when thermal radiation is applied to microphysics. The dynamical feedback is shown to be an enhanced cloud circulation with stronger subsiding shells at the cloud edges due to thermal cooling, and stronger updraft velocities in the cloud center. It is shown that an evaporation-circulation feedback reduces the amount of rain produced in simulations where 3D thermal radiation is applied to microphysics and dynamics, in comparison where 3D thermal radiation is only applied to dynamics.

## 1 Introduction

Cloud droplets form in saturated environments by condensation of water vapor on cloud condensation nuclei (CCN). In the first phase of its lifetime, cloud droplet growth follows Köhler theory (Köhler, 1936). If a certain critical radius is reached a droplet can grow further, following diffusional droplet growth theory. From a certain droplet size onward, rain formation processes such as collision and coalescence dominate growth (Pruppacher and Klett, 2010).

The droplet size distribution in clouds has important implications for the Earth's atmosphere. The size distribution of droplets

determines how much solar radiation is reflected back to space. Smaller droplet sizes reflect more radiation back to space (for constant liquid water), thus leading to a cooling of the atmosphere while larger droplets allow radiation to penetrate more easily to the surface, thus allowing more radiation to be absorbed (Ramanathan et al., 1989; Stephens, 2005; Boucher et al., 2013). Furthermore, the droplet size distribution determines the formation of rain in clouds. Droplets that reach the 10 - 30 $\mu$m radius range can lead to rain formation. Only very small numbers of droplets of this size (order 1 per liter) are necessary to initiate the process of collision and coalescence. It is known that a broad droplet size spectrum is necessary for these processes to start, however cloud droplet growth in the diffusional growth theory slows down when droplets reach 10 $\mu$m and collision and coalescence is not yet effective (the so called collision-coalescence bottleneck Simpson (1941); Langmuir (1948); Mason (1960); Pruppacher and Klett (2010)). Different processes can cause broadening of the droplet size spectra, e.g. turbulence (Grabowski and Wang, 2013), the associated supersaturation fluctuations (Cooper, 1989; Shaw et al., 1998; Lasher-Trapp et al., 2005; Grabowski and Abade, 2017), giant CCN (Feingold et al., 1999; Cheng et al., 2009) and radiation (Harrington et al., 2000; de Lozar and Muessle, 2016). As soon as rain is initiated, the cloud system morphology and intrinsic properties can change as the dynamics of the system change.

Radiative effects on cloud droplet growth have been studied in various ways in the past. Among the earliest are the studies by Roach (1976) and Barkstrom (1978). Both analyzed the growth of an individual droplet and showed that droplets can grow to 20 $\mu$m and larger by radiative cooling, even in a subsaturated environment. Guzzi and Rizzi (1980) and Austin et al. (1995) studied the effect of radiation on the growth of a droplet population. Guzzi and Rizzi (1980) showed increased droplet growth in the diffusional droplet growth regime, while Austin et al. (1995) also included collision-coalescence and found earlier onset of rain by a factor of 4. An important issue of the application of radiation to droplet growth is the time scale of the temperature exchange. Davies (1985) estimated the time until droplets reach a steady state in temperature exchange. For most droplet sizes, the time scale was small enough to make the assumption of a steady state system feasible. Bott et al. (1990) simulated radiative fog, thus including microphysical and dynamical feedbacks. The inclusion of the radiative term in the droplet growth equation had important consequences for the lifetime of fog. The enhanced growth of larger droplets by radiation and associated gravitational settling caused a reduction of liquid water in the fog. The oscillation of liquid water (with a period of 15-20 min) could only be simulated by including radiative effects. Ackerman et al. (1995) simulated stratocumulus clouds, using a bin microphysical model including 1D radiation. The stronger diffusional growth in the simulations with radiative effects reduced supersaturation and therefore the number of small droplets. The reduced number of droplets and the larger droplet size resulted in more drizzle and therefore a lower cloud optical thickness. Observations of nocturnal stratocumulus were remodeled with Lagrangian parcels by Caughey and Kitchen (1984). They stated that: 'A simple Lagrangian model suggested that the larger drops grew within the zone of high net radiative loss around cloud top'. Harrington et al. (2000) used a large eddy simulation (LES) and an independent parcel model, including bin microphysics and radiative effects on droplet growth. They showed that only parcel trajectories spending long periods of time at cloud top (10 minutes or more), can cause the droplet size spectrum to broaden via radiative cooling. They also found an earlier onset of drizzle production, however, this occurred along parcels that would produce drizzle anyhow. They concluded that radiative cooling may reduce the time for drizzle onset. The recent theoretical study of Brewster (2015) and direct numerical simulations by de Lozar and Muessle (2016) re-emphasize the hypothesis

that thermal radiation might influence droplet growth significantly and lead to a broadening of the droplet size spectra and thus enhance the formation of precipitation. Similarly, Zeng (2018) investigated the effect of thermal radiation on rain formation in a precipitating shallow cumulus case and found broadening of the droplet size spectrum and earlier rain formation.

In this study, we investigate the role of thermal radiation on cloud droplet growth in cumulus clouds. The limited lifetime of cumulus clouds changes the radiative impact compared to former studies where stratiform clouds where investigated. The finite size of the cumulus clouds and the high local cooling rates of several hundred K/d at cloud top and at cloud sides (e.g. Kablick et al. (2011); Klinger and Mayer (2014); Klinger et al. (2017)) suggest that the investigation of 3D thermal radiation effects might have a significant effect on drop growth. Klinger and Mayer (2016) (their Figure 11) showed that local peak differences in cooling rates between 1D and 3D thermal radiation in cumulus cloud fields can reach 20 - 120%, depending on the cloud field resolution. But the differences between 1D and 3D thermal radiation are not only focused on local grid boxes. Kablick et al. (2011) and Črnivec and Mayer (2019) showed that layer averaged 1D and 3D heating and cooling differences can be up to 1 K/d, which is the same order of magnitude as clear sky cooling. Whether the stronger local 3D cooling affects droplet growth compared to 1D thermal cooling and, if cooling in general causes changes in droplet growth in cumulus clouds, are questions addressed in this study. The focus of this work is on thermal radiative effects on droplet growth. At the end of the study, we will briefly investigate thermal radiative effects on dynamics as e.g. shown by Guan et al. (1995, 1997), Mechem et al. (2008) or Klinger et al. (2017).

The paper is structured as follows: Section 2 provides the necessary theory, section 3 the model setup. Section 4 analyses the results of our study. Summary, conclusion and outlook are provided in Section 5.

## 2 Theory

The energy budget at a droplet surface is described by Eq.( 1) which combines water vapor diffusion to the droplet and latent heat release, where $l_v$ is the latent heat, $dmdt^{-1}$ the change in mass $(m)$ over time $(t)$, $r$ the droplet radius, $K$ the thermal diffusivity and $T_{\mathrm{d}}$ and $T_{\mathrm{inf}}$ the droplet temperature and the temperature of the surrounding air:

$$l_v \frac{dm}{dt} = 4\,\pi\,r\,K\,(T_d - T_{\mathrm{inf}}). \tag{1}$$

Following Roach (1976), the equation can be extended by a radiative term (Eq. (2)), where $HR_\lambda(r) = 4\pi r^2 q_{abs,\lambda}(r) F_{net,\lambda}(r)$ is the emitted or absorbed power of an individual droplet. $q_{abs,\lambda}(r)$ is the absorption efficiency per droplet radius and wavelength $(\lambda)$ and $F_{net,\lambda}(r)$ the net radiative gain/loss of a droplet per radius and wavelength in W/m$^2$.

$$l_v \frac{dm}{dt} = 4\,\pi\,r\,K\,(T_d - T_{\mathrm{inf}}) + \int_\lambda HR_\lambda(r)\,d\lambda. \tag{2}$$

Harrington et al. (2000) transformed the equation to the notation of the bin microphysical model of Tzivion et al. (1989). We will follow their notation in the following, as we use the same bin microphysical model. Thus, Eq. (2) becomes

$$\frac{dm}{dt} = C(P,T)\,\frac{m^{2/3}}{m^{1/3}+l_0}\left[\eta(t) + J(P,T)\,m^{1/3}\,HR(m)\right] \tag{3}$$

where $l_0$ is a length scale representing gas kinetic effects, $\eta(t)$ the excess specific humidity ($q_v$-$q_s(T)$) and $C(P,T) = \frac{4\pi}{Cr_s}$; $C = \frac{R_v T_{\text{inf}}}{De_s(T_{\text{inf}})} + \frac{l_v}{T_{\text{inf}}K}\left(\frac{l_v}{R_v T_{\text{inf}}} - 1\right)$ where $D$ is the diffusion coefficient, $R_v$ is the specific gas constant for moist air and $e_s$ is the saturation vapor pressure. $J(P,T)$ summarizes constants concerning the radiative term: $J(P,T) = \frac{r_s l_v \alpha_c}{K R_v T}$ with $r_s$ the saturation mixing ratio, $\alpha_c = \left[\frac{3}{4\pi\rho_l}\right]^{\frac{1}{3}}$, and $\rho_l$ the liquid water density. We note that the radiative cooling is an increasing function of droplet mass.

$HR(m)$ of a droplet is the wavelength band ($i$) integrated radiative gain or loss, weighted by the absorption efficiency for a mass size-bin ($k$) for the bin microphysical model. Harrington et al. (2000) showed that the radiative term $HR(m)$ can be approximated with the mean mass ($\overline{m}_k$) of a drop size bin $k$:

$$HR(m) = \sum_{i}^{N_{bands}} q_{abs,i}(m)\, F_{net,i} \approx$$

$$\sum_{i}^{N_{bands}} \overline{q}_{abs,i}(\overline{m}_k)\, F_{net,i} = HR(\overline{m}_k). \tag{4}$$

This radiative term must be included in the equation for supersaturation and for droplet growth. The equation for the supersaturation, in our case water vapor excess $\eta$, is:

$$\frac{d\eta}{dt} = D - A(P,T)\frac{dM}{dt} \tag{5}$$

where the function $A(P,T)$ connects the integrated mass growth rate $dMdt^{-1}$ to changes in $\eta$. Including the integrated radiative terms of the mass growth rate $\mathcal{R}$, Eq. 5 becomes

$$\eta(t) = \left\{\left[\eta(t_0) - \frac{D}{G}\right]e^{-G(t-t_0)} + \frac{D}{G}\right\} - \frac{\mathcal{R}}{G}\left[1 - e^{-G(t-t_0)}\right] \tag{6}$$

where $D$ represents the increase/decrease in $\eta$ due to dynamics, $G$ is the contribution to $\eta$ from the standard droplet growth and $\mathcal{R}$ the contribution to $\eta$ from radiatively driven droplet growth. Here it can be seen that the additional radiative term can increase/decrease $\eta$ due to radiative heating/cooling. For a more detailed explanation the reader is referred to Harrington et al. (2000), their Equations 6 - 10.

For solving the condensation equation in the two-moment framework of Tzivion et al. (1989), where both mass and number in a bin $k$ are predicted, Eq. (3) has to be integrated over one time step from $t_0$ until $t_f$. Again, we follow Harrington et al. (2000) to calculate the forcing $\tau$ (the gain or loss of mass of a droplet) of the droplet growth equation:

$$\int_{m_o}^{m_f} \frac{m^{1/3} + l_o}{m^{2/3}} =$$

$$C(P,T)\int_{t_0}^{t_f} \eta(t)\, dt + \overline{m}_k^{(1/3)}\, C(P,T)\, J(P,T)\, HR(\overline{m}_k)\, \Delta t =$$

$$\tau_d + \tau_r = \tau \tag{7}$$

where $\tau$ is the combined dynamic ($\tau_d$) and radiative ($\tau_r$) forcing of the droplet growth equation and $m_0$ and $m_f$ are the initial and final mass of the droplet before and after condensation/evaporation.

What remains now is to derive the radiative term $F_{net,\lambda}(r)$ in Eq. 2 from the radiation scheme in the LES model. Heating rates in LES models are calculated spectrally from bulk water. These heating rates include contributions from liquid water/cloud water as well as water vapor and other atmospheric gases. Former studies, e.g. Roach (1976) and Harrington et al. (2000), used a 1D radiative transfer approximation and calculated the individual droplet absorption and emission from the upwelling and downwelling fluxes. We, however, include 3D-radiative effects. Our 3D radiative transfer approximation is designed to provide 3D heating rates. We estimate the individual droplet emission/absorption from a volume heating rate and therefore have to separate the heating/cooling from the liquid water phase ($HR_{liquid}$) from the total heating/cooling (from liquid water and atmospheric gases, $HR_{tot}$). We follow the approach of Mayer and Madronich (2004) which showed the relationship between heating/cooling rates and the actinic flux $F_0$ to be

$$HR_{tot,\lambda} = -k_{abs,\lambda}F_0$$
$$HR_{liquid,\lambda} = -k_{abs,liquid,\lambda}F_0 \tag{8}$$

where $k_{abs}$ is the total absorption coefficient and $k_{abs,liquid}$ the absorption coefficient of liquid water.

Combining these two equations it follows that the heating/cooling rate resulting from the liquid water absorption is

$$HR_{liquid,\lambda} = -\frac{k_{abs,liquid,\lambda}}{k_{abs,\lambda}}HR_{tot,\lambda}. \tag{9}$$

This total heating rate now has to be distributed among all droplets in the volume. The total heating or cooling from the liquid water of a grid box (for a single wavelength $\lambda$ or wavelength band $i$) is the sum of all droplet contributions to the heating or cooling

$$HR_{liquid,\lambda} = \int n(r)h_\lambda(r)dr \tag{10}$$

where $n(r)$ is the number of droplets of radius $r$ per radius interval $dr$ and $h_\lambda(r) = 4\pi r^2 q_{abs,\lambda}(r)F_{net,\lambda}(r)$ the heating or cooling rate of each droplet at radius $r$ with the absorption efficiency $q_{abs,\lambda}(r)$ and the net-heating of each droplet $F_{net,\lambda}(r)$.

Assuming steady state (e.g. Davies (1985)), $HR_{liquid,\lambda}$ is equally distributed among all droplets and the individual heating or cooling ($F_{net,\lambda}(r)$) of a droplet of size $r$ in Eq. (2) is therefore

$$F_{net,\lambda}(r) = \frac{HR_{liquid,\lambda}}{\int 4\pi r^2 n(r)q_{abs,\lambda}(r)dr}. \tag{11}$$

## 3  Methodology

To estimate the effect of 1D and 3D thermal radiation on cloud droplet growth we use a two-stage approach. First, to estimate the impact of thermal radiation on droplet growth and to gain insight into physical processes, we use Lagrangian parcels (Yamaguchi and Randall, 2012) recorded during a LES simulation (System for Atmospheric Modelling, SAM; Khairoutdinov and Randall (2003)) with the bin emulating two moment bulk scheme of Feingold et al. (1998b). These parcel trajectories are then used to drive an independent (offline) parcel model including a bin microphysics scheme (Tzivion et al., 1987; Feingold et al., 1999). We separate between 1D (RRTMG, Mlawer et al. (1997); Iacono et al. (2000), 1DR) and 3D thermal (Klinger and Mayer (2016), 3DR) radiative effects and compare both results to the droplet growth without radiative impacts (NR) and to each other. This approach allows us to focus on the effect of thermal radiation on droplet growth, without the interaction of changing dynamics that would occur in a fully coupled LES simulation.

Second, we run a fully coupled LES simulation with the TAU bin-microphysics scheme (Tzivion et al., 1987; Feingold et al., 1988; Tzivion et al., 1989), where 1D and 3D thermal radiative effects (heating rates) are applied to the droplet growth, and to the dynamics, or to just one of the two. We chose a shallow cumulus case with weak precipitation (BOMEX) where we expect the effects of thermal radiation on cloud droplet growth to be tangible, and not overwhelmed by the rapid development of precipitation encountered in deeper trade-wind cumulus environments. We expect that it would be harder to discern these effects in a more strongly precipitating case.

In both cases the simulations were run with 75 m horizontal and 50 m vertical resolution for a 45x45 km$^2$ domain. The simulations for the trajectories were run for 6 hours in total; the last 2 hours are used for evaluation. The coupled LES cases were run for 8 hours in total.

For the first part of the study 2.7 million Lagrangian air parcel trajectories were recorded in the last two hours of the BOMEX simulation with a 2 second time step. The simulation was driven by 1D thermal radiation, but we recorded 3D thermal radiation along the same parcels. This allows us to compare the same parcels, driven by the same variables (liquid water potential temperature, pressure, vertical velocity) in the later part of the study. The difference in the results of the independent parcel model ensemble is therefore only due to the difference in the 1D and 3D thermal heating/cooling rates and their impact on cloud droplet growth. (Changes to the approach of Harrington et al. (2000) were explained in Section 2.) The total number of aerosol particles (assumed to be ammonium sulfate) is 100 cm$^{-3}$ with a median radius of 0.1 $\mu$m and a geometric standard deviation of 1.5 (assuming a log-normal distribution). The bin model includes diffusional growth and the growth by collision and coalescence and covers 33 size bins with a mass doubling from one bin to the next. The radius of the first bin (lower bound) is 1.56 $\mu$m. Aerosol particles are activated based on the locally calculated supersaturation, and placed in the first bin. We neglect solute- and kelvin-effect in this framework, because they have a minor impact for $r > 1.56$ $\mu$m. Kinetic and ventilation effects are taken into account.

A few comments are in order regarding our approach. With the parcel model, we focus on the effects of thermal radiation on microphysics, neglecting any changes in cloud development that would occur due to feedbacks within a LES framework. A further advantage of this method is that spurious spectral broadening due to advection is avoided (Harrington et al., 2000). A

key limitation of this method is that drop sedimentation is not represented; drops do not fall off a trajectory of interest, and drops from other trajectories do not fall onto that trajectory. Because all droplets follow the parcel trajectory, the liquid water content ($q_c$) is not reduced as the parcels do not 'rain out' and radiation does not change along the parcel trajectories when the size distribution (or $q_c$) changes. The method is thus mostly useful for examining the *onset* of drizzle. One can consider the trajectory approach to be an imperfect but useful model (as documented in Stevens et al. (1996), Feingold et al. (1998a) and Feingold et al. (1999)) for examining the combined effect of droplet growth and thermal radiation with and without the radiative effects in a framework that allows for realistic and quantifiable exposure to strong radiative cooling at cloud edges. The analysis of characteristic timescales of important processes for a droplet radius of 20 $\mu m$, such as diffusional droplet growth ($\chi_{growth}$), diffusional droplet growth with radiation ($\chi_{growth,rad}$), and sedimentation ($\chi_{sed}$) supports our argument about the usefulness of the approach, despite the fact that sedimentation is not represented in the parcel model. The characteristic timescales for the three processes are on the order of minutes for the diffusional dropelt growth ($\chi_{growth} = 6\ min\ 40\ s$ and $\chi_{growth,rad} = 5\ min\ 30\ s$), and on the order of an hour for sedimentation ($\chi_{sed} = 1\ h\ 23\ min$). For the full calculation of the timescales, the reader is referred to the appendix (Appendix A). This clear signal lends credence to the use of the parcel model.

In contrast, LES allows for for a more faithful treatment of these processes because of the coupling of interactive components, but at the expense of transparency of the radiative effects on drop growth. In combination the two modeling approaches allow insights that neither could have produced by themselves.

As the microphysical schemes in our LES simulation and in the offline parcel model are different (2 moment bulk vs. bin), small differences in the predicted liquid water can occur. Therefore, the calculated heating/cooling rates of the LES might occasionally be too high for the application in the parcel model, thus causing unrealistic droplet growth. We therefore applied a threshold to the cooling. Whenever the distributed droplet cooling ($F_{net}$) was larger than the black body emission ($\sigma T^4/6$; the factor 1/6 accounting for the window regions and emission to only one hemispheric dimension) the cooling of the droplet was set to the black body emission value. Tests showed that the discrepancy between the liquid water content of the parcel model and the LES occurs most often at the edges of clouds where $q_c$ is very small. In this area, droplet cooling can be regarded as "black", because droplets are exposed to clear sky.

The coupled LES simulation have a similar setup, but we used the bin-microphysics scheme from the beginning of the simulation. We restarted after 4 hours from a base simulation with 1D thermal radiation passed to dynamics only. We separate 5 cases:

- 1D thermal radiation applied to dynamics only ($1DD$)

- 1D thermal radiation applied to dynamics and droplet growth ($1DD\_1DM$)

- 3D thermal radiation applied to dynamics only ($3DD$)

- 3D thermal radiation applied to dynamics and droplet growth ($3DD\_3DM$)

- 1D thermal radiation applied to dynamics and 3D radiation applied to droplet growth ($1DD\_3DM$)

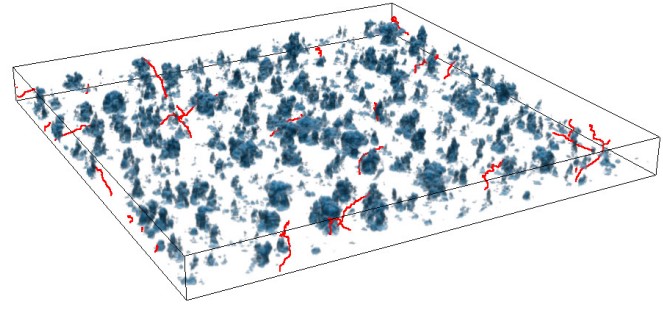

**Figure 1.** Time snapshot of the BOMEX shallow cumulus cloud field. Displayed is $q_c$ and selected parcel trajectories (red).

These five simulations allow us to a) look at the effect of thermal radiation on droplet growth, b) separate between 1D and 3D thermal radiative effects and c) to separate the droplet growth effect from dynamical effects.

## 4 Simulations Results

### 4.1 Parcel Model - Cloud Field Statistics and Properties

Figure 1 shows a time snapshot of the cumulus field and selected time-dependent trajectories (red). From our 2.7 million parcel trajectories we selected about 340,000 that make contact with a cloud for further investigation. This number was chosen as it provides us with a statistically representative result, and a number of parcels that could still be handled in a finite amount of time in the post processing.

The effect of 1D and 3D thermal radiation on the growth of cloud droplets depends (among other factors) on the length of time
that a droplet is exposed to thermal cooling (in other words, that a droplet is located close to cloud edges/cloud top) and the strength of the cooling. Harrington et al. (2000) found that droplets have to spend about 10 min in a cooling area to experience a noticeable effect on the droplet size distribution. We therefore first investigated different properties of our trajectories:

- In-cloud residence time

- Time spent in the vicinity of cloud edges/tops

For the cloud residence time, we used a threshold of 0.01 gkg$^{-1}$ to separate between cloudy and cloud free areas. We then traced among our 340,000 parcel the time periods during which a parcel stays in a cloud. Inevitably, a parcel can contact a cloud more than once, in which case, the hits were counted as multiple events. Figure 2 shows a histogram of the time that our parcels spend in clouds. Most of the parcels spend less than 15 min in a cloud, but we also find some rather long periods of more than 25 min. This is in agreement with former results (e.g. Jiang et al. (2010)).

The time at cloud side was estimated by a) setting the same threshold for $q_c$ (0.01 gkg$^{-1}$) and additionally setting four different thresholds in terms of heating rates (-4 Kd$^{-1}$, -10 Kd$^{-1}$, -20 Kd$^{-1}$, -100 Kd$^{-1}$) for 1D and 3D thermal radiation. Again, multiple hits were possible for each parcel trajectory. The histograms are shown in Fig. 3. 1D (blue) and 3D (orange) thermal

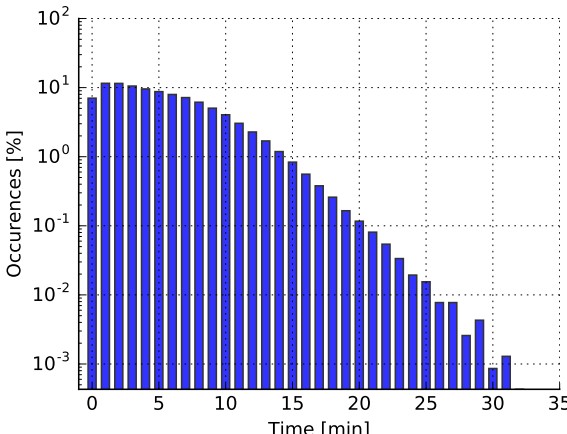

**Figure 2.** Histogram of the time that parcels spend in a cloud. For the sampling of the data, a threshold of 0.01 g kg$^{-1}$ of the $q_c$ was used to separate cloudy from non-cloudy regions.

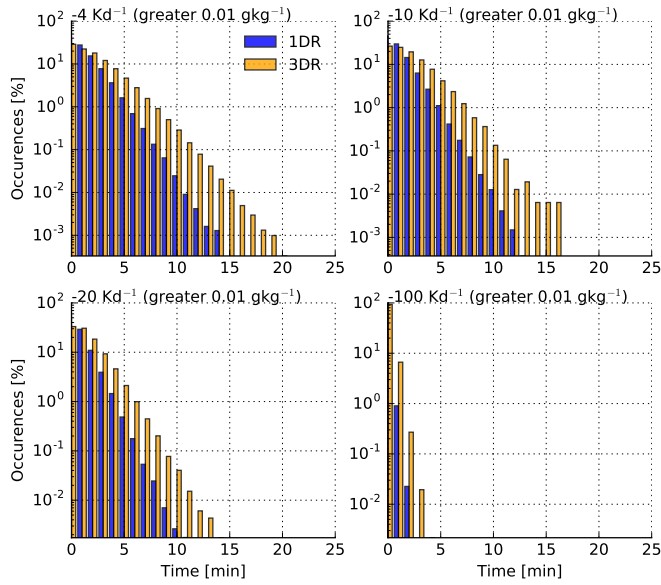

**Figure 3.** Histogram of the time that parcels spend at cloud top or cloud side. For the sampling of the data, a threshold of 0.01 gkg$^{-1}$ of the $q_c$ was used to separate cloudy from non-cloudy regions. To separate cloud edge regions from the cloud interior, 4 different thresholds of the cooling rates were used (4 Kd$^{-1}$, 10 Kd$^{-1}$, 20 Kd$^{-1}$, 100 Kd$^{-1}$).

radiative transfer simulations show that most of the parcels spend less than 5 min in a certain cloud volume encompassing a cooling threshold. For the 100 Kd$^{-1}$ threshold, no parcel exceeds 3 minutes. However, there are some parcels which spend

10 min or longer, especially when 3D radiative effects are considered in volumes experiencing cooling of 10-20 $Kd^{-1}$. This is simply due to the fact that a larger volume of each cloud experienced cooling rates in 3D radiative transfer. The possibility that thermal radiation can affect cloud droplet growth is therefore given. In the following, we will take a closer look at individual parcel trajectories and the overall statistics of the 340,000 parcel trajectory ensemble.

## 4.2 Parcel Model - Cloud Droplet Growth including Thermal Radiative Effects

### 4.2.1 Individual Parcels

We now focus on individual parcels. An example of a parcel trajectory is given in Fig. 4. The $q_c$ is shown in color (Fig. 4 a). This illustration of the trajectory includes a temporal dimension. Each data point is recorded at a different time step. The parcel rises in the beginning, enters an area of high $q_c$ (red, arrow i)), followed by a decrease in $q_c$ (blue area, arrow ii)), but never drops to zero, before entering again an area of high $q_c$ (red, arrow iii)). Finally, $q_c$ decreases again and the parcel leaves the cloud.

The other three panels of Fig. 4 combine time snapshots of the cloud field and the temporal development of the parcel trajectory. The cloud field is shown at the time marked by the red dot on the trajectory. The surface shows the liquid water path, $lwp$, of the selected cloud field at that specific time. The red dot on the surface is the vertical projection of the location of the parcel at the time. Figure 4 b) shows the updraft area where the parcel first enters an area of high $q_c$. The parcel (at that time) is located in the upper part of the cloud where it experiences cooling. In the following, the cloud grows and at time c) a significantly larger cloud with more $q_c$ is encountered. The parcel is now located at the outer edge of the cloud (especially visible in the $lwp$ field, red dot). $q_c$ has dropped below 0.01 $gkg^{-1}$, but does not decrease to zero in the following, meaning the parcel never leaves the cloud. The cloud grows further (Fig. 4 d)) and the parcel is located again in an area of high $q_c$. We will see later that this 'recirculation' of parcels occurs occasionally and can cause a broadening in the droplet size spectrum. It is likely that radiative effects become more important in this case, because parcels pass cloud edges where thermal cooling per droplet is strong.

Parcel Trajectory 1

We now take a more detailed look at the same parcel trajectory shown in Fig. 4. Figure 5 shows this selected parcel, which is characterized by moderate vertical velocities (peaking at about 6 $ms^{-1}$ in the beginning, but not exceeding 2 $ms^{-1}$ later on). The parcel stays in the cloud for about 20 minutes and twice experiences radiative cooling (for about 8 minutes and 2 minutes). We chose four different time steps for further investigation (red dotted lines at 14, 16, 21 and 25 minutes). The first time step was chosen shortly after the parcel passes the first volume of strong cooling and is recirculating. Here, we defined 'recirculation' loosely as an event where the $q_c$ along a parcel trajectory becomes very low (in this case 0.007 $gkg^{-1}$). The second time step was chosen after $q_c$ has risen again, the third time step shortly before the second cooling phase, and the fourth when the parcel leaves the cloud.

The drop distribution at these four time steps is shown in Fig. 6. The upper figures show the drop size spectra (dm/dr) them-

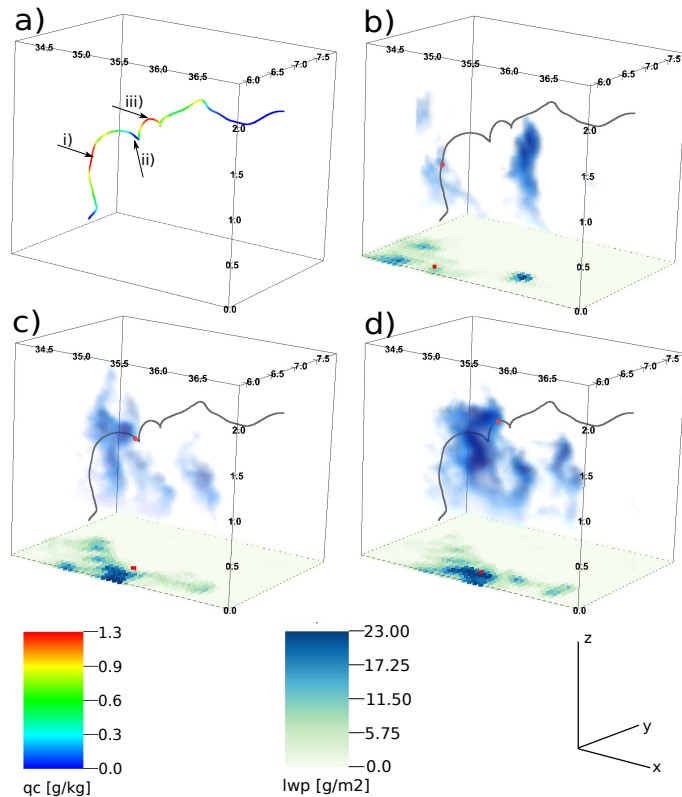

**Figure 4.** 3D Visualization of $q_c$, the parcel position and $lwp$ of the selected scene. The first figure (upper left) shows the parcel trajectory. The $q_c$ at each time step of the selected parcel is colored. Three time intervals were selected for the following figures: time intervals where the parcel stays in high $q_c$ areas (i,iii) and one, where the $q_c$ drops substantially but does not reduce to zero (i). The following 3 figures show the parcel trajectory (gray, again time dependent). For the time interval in focus, the $q_c$ is again colored. The displayed clouds are chosen at the center time of the interval, as is the $lwp$. The red marker displayed in the $lwp$ field shows the projected location of the center time step.

selves, the lower figures the ratio of the spectra of the 1DR/3DR simulations and the NR simulation of the parcel.

In the beginning, hardly any differences can be seen in the drop spectra between the NR, 1DR and 3DR simulations. The spectrum broadens over time. Looking at the ratio of dm/dr of the 1DR/3DR and the NR simulation reveals a decrease in mass in the small bins and an increase in the larger bins for the radiation simulations. This changes later in the simulation when the simulations with thermal radiation increase mass over the entire drop spectrum. dm/dr for the 1DR and 3DR simulations exceeds the NR simulations up to 1.5 times for $r > 10~\mu$m. The size spectrum broadens and a drizzle mode develops. The peak of the spectra remains at about 15 $\mu$m. The ratio for 3DR simulations always exceed that for 1DR simulations. A factor of more than 2 is reached for dm/dr of the 3D radiation simulation compared to the NR simulation. The droplet concentration in the 20 $\mu$m bin (not shown) increases by up to 15% for the 3DR case along this trajectory.

The possible increase in droplet growth by thermal radiation does not only depend on the time that a droplet is exposed to cooling, but also on the magnitude of the cooling and the size of the droplet. Recall that the larger the droplet, the more ef-

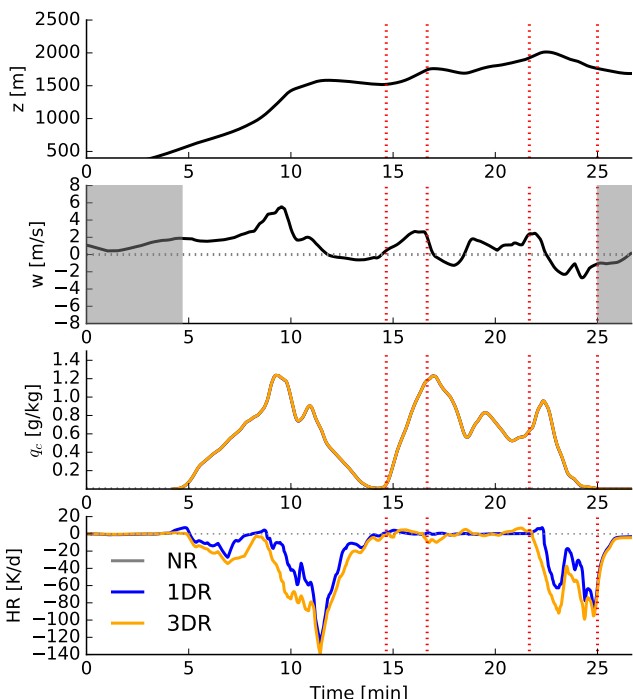

**Figure 5.** Time series of different properties of the first selected parcel. Shown are: height, vertical velocity, $q_c$, heating/cooling rate, predominant radius and the heating/cooling rate per droplet. Gray areas show time intervals where the $q_c$ is below 0.01 $\text{gkg}^{-1}$. The red dotted lines show selected time steps used in the following analysis.

fectively radiation can act on it as the droplet absorbs and emits radiation more effectively. Radiative effects become stronger from a radius of about 10 $\mu$m on. Additionally, the radiative impact competes with the dynamical effects, which depend on the vertical velocity (see Eq. (7)). It therefore follows that the larger the droplet and the weaker the updraft, the more radiation can affect droplet growth. Figure 7 shows the temporal development of the individual droplet heating/cooling rate for three

5     different sizes (left column). This heating/cooling rate is the fraction of the spectrally- and bin-integrated cooling rate per bin (Eq. (4)), integrated over all wavelengths. The center radius of the corresponding bin is given in each figure. Gray shaded areas and the red lines are identical to those shown in Fig. 5 for comparison. Note the change in the y-axis for these figures. We find that the cooling per droplet increases with increasing radius and that 3D cooling is stronger than 1D cooling.

    The right column shows the forcing $\tau$ (Eq. (7)), which is the total driving force for condensation in each bin. The gray line

10    shows the dynamical forcing ($\tau_d$), which, if compared to Fig. 5 follows the vertical velocity trend. The radiative forcing ($\tau_r$) is shown in yellow (3D) and blue (1D) for the same four size bins. Note that a cooling per droplet (left side) causes a positive contribution to the droplet growth and therefore a positive forcing (right side). The radiative forcing is smaller than the dynam-

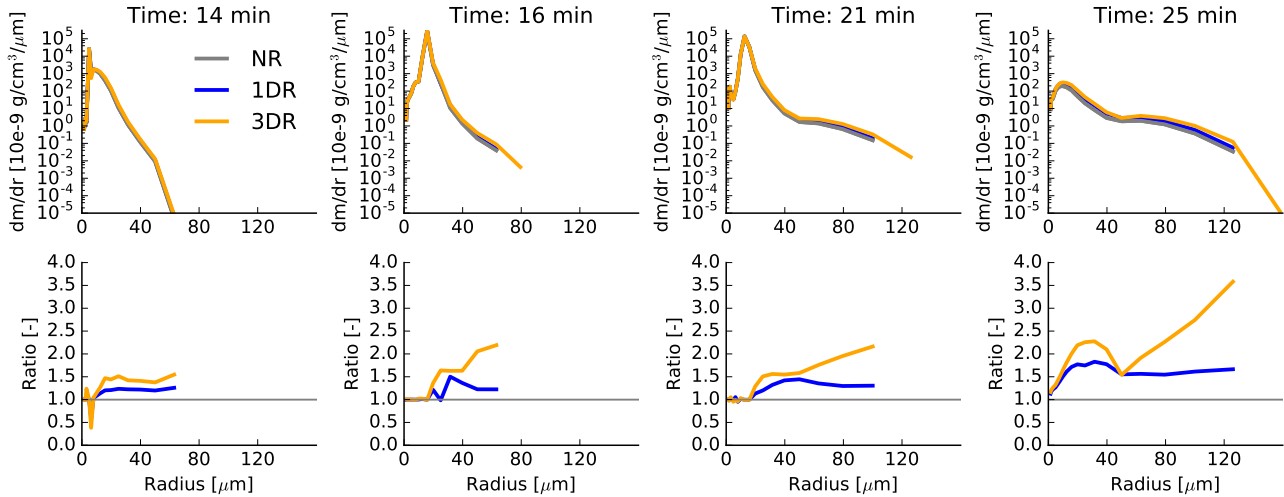

**Figure 6.** Drop size distribution dm/dr plot for the four selected time steps displayed by the red dotted line in Fig.5. The upper row shows dm/dr. The lower row shows the ratio between the NR case and the 1DR/3DR case. Gray areas in the lower row display size bins where mass occurs in the 1DR/3DR case, but not the 1DR case and wherefore no ratio could be calculated.

ical one but has the same order of magnitude. An additional boost is given to the droplet growth shortly before 15 min, after which the dynamical forcing rises again. This small radiative perturbation is sufficient to cause the increase in dm/dr seen in the second column of Fig. 6. The radiative forcing becomes strongest towards the end of the parcel trajectory, counteracting the negative dynamical forcing, especially for the larger size bins.

### Parcel Trajectory 2

This second parcel experiences stronger dynamical forcing. Vertical velocity rises and falls throughout the parcel's lifetime and peaks at more than +/- 6 ms$^{-1}$. The parcel recirculates twice. During these two periods the parcel experiences radiative cooling, which causes a broadening of the droplet size spectrum (Fig. 8 and Fig. 9). Due to the strong dynamical forcing, the

10 parcel shows broader spectra than the first trajectory. As before, a substantial increase in condensed water is found for the 1DR and 3DR simulations, peaking for the 3D thermal radiation simulation.

These examples illustrate the variety of ways in which radiative effects can act on a cloud droplet. The time that a parcel spends in a certain cooling area, the magnitude of the cooling, the size of the droplets at the time of cooling and the dynamical forcing contribute to droplet growth with different magnitudes. The ideal situation for the radiative effects to enhance droplet

15 growth would be droplets of size of about 10 $\mu m$ or more, a cooling period of more than 5 minutes in a cooling of 20 Kd$^{-1}$ or more, and vertical velocities close to zero. Because these effects usually do not occur together, the overall effect on the droplet growth from these factors is small.

A strong effect is found when parcels recirculate. Whenever a parcel reaches cloud edge, the number of droplets is small.

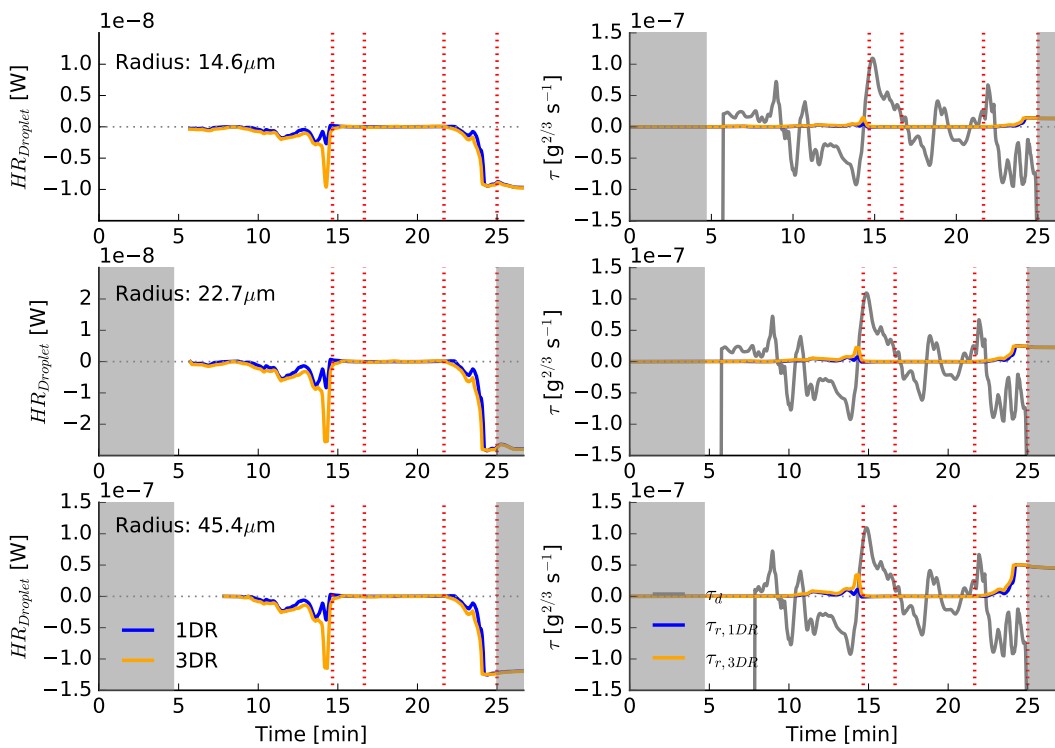

**Figure 7.** Wavelength integrated, bin-resolved heating/cooling rates and forcing $\tau_d$ and $\tau_r$

Yet, these droplets are exposed to cloud top/ cloud edge cooling which, due to the limited number of droplets is close to the maximum cooling that a droplet can experience (2 in Fig. 10). Additionally, these parcels already include larger droplets, where radiative effects are stronger. The droplets experience additional growth by radiative cooling during the recirculation time and return into the cloud with a slightly broadened size distribution (3 in Fig. 10). The droplet size distribution subsequently
5    continues to broaden (4 in Fig. 10).

Summarizing, these analyses of individual parcel trajectories have shown that radiatively enhanced droplet growth can occur in 'lucky situations' or when recirculation occurs. The increased droplet growth for recirculating parcels agrees well with prior results of enhanced droplet growth in areas of net radiative loss (see e.g. Caughey and Kitchen (1984)). The radiative cooling does not seem to cause droplet growth in individual parcels beyond the NR case (as also found by Harrington et al. (2000)), but
10    thermal radiative effects enhance the mass per bin and occasionally allow droplets to grow into larger bins. In the following, we will take a more general look at the effects in our parcel trajectory ensemble.

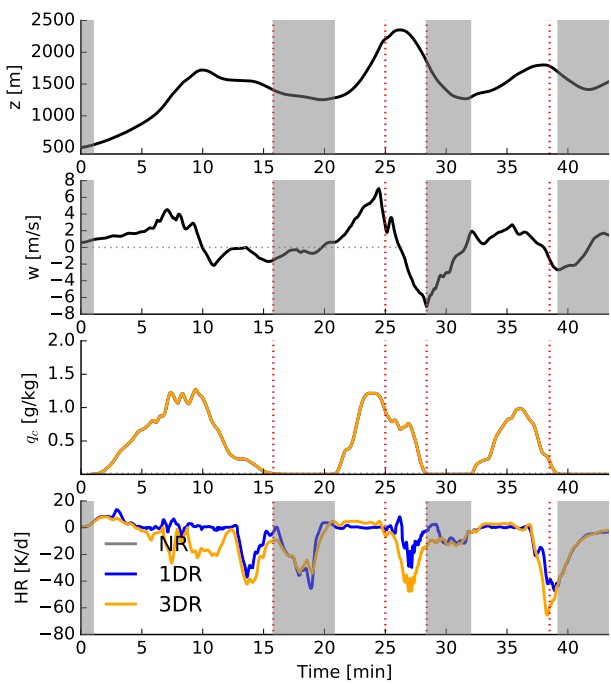

**Figure 8.** Similar to Fig. 5, but for the second selected parcel.

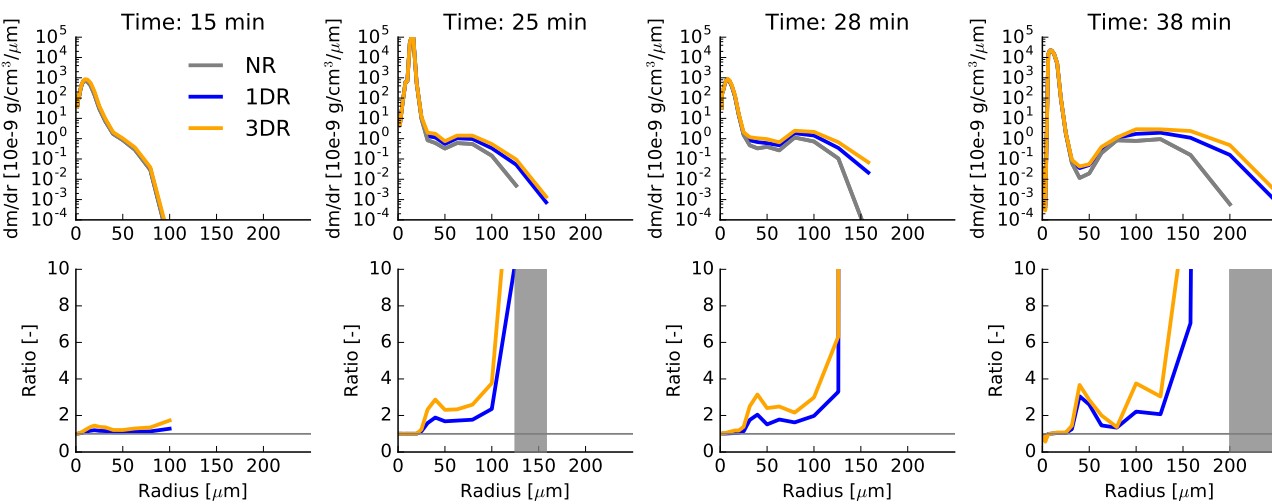

**Figure 9.** Similar to Fig. 6, but for the second selected parcel.

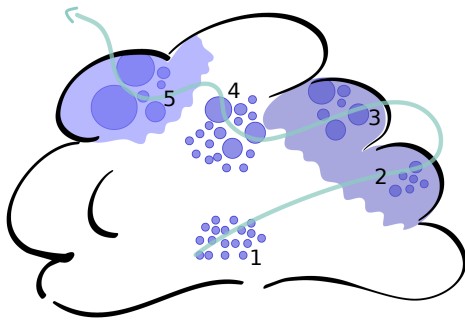

**Figure 10.** Schematic figure of the droplet growth for a recirculating parcel.

### 4.2.2 Parcel Model - Ensemble Results

As a next step, we evaluated our 340,000 parcel trajectory ensemble to see if we find changes in droplet size and rain amount, as represented by the the local volume flux of water, or rain rate.

Figure 11 shows a histogram of maximum mean radius along a parcel trajectory versus the integrated $q_c$ along the parcels. The first figure shows the number of occurrences for the NR case. Integrated $q_c$ mostly occurs in a range of 0 - 10 g kg$^{-1}$ min with maximum mean radii up to 20 $\mu$m. Larger droplets and integrated $q_c$ amounts exist, but are comparatively small in number. When comparing the number of occurrences of the 1DR case to the NR case, we find an increase in the number of larger droplets for small $q_c$ amounts and for those between 5 to 10 g kg$^{-1}$ min and a decrease in the directly smaller bin. Radiation thus enhances the growth of droplets for a specific $q_c$ for very small droplets and for droplets in the 10 - 25/30 $\mu$m range. There is also a tendency for the larger drops to grow to larger sizes in the 1DR case. For 3D thermal radiation we see a similar picture. The number of droplets growing to larger sizes is even higher than in the 1DR case. Comparing the results of the 3D thermal radiative transfer simulation to the 1DR simulation shows the additional increase in the 3DR case. We confirm here that due to thermal radiation droplets in the critical range tend to grow to larger sizes.

Next, we calculate the total rain rate at each time step (accounting for drop radii $> 20$ $\mu$m) for the entire trajectory ensemble. In the first hour, the absolute differences in the rain rate between the three setups is small. Absolute differences become larger over time and are clearly visible during the last 40 min of the analyzed time period (Fig. 12). Looking at the relative differences between either the 1DR simulation and the NR case, or 3D thermal radiation simulation and the NR case, we find differences of 10% for the 1DR case in the first hour, and 20% for the 3D thermal radiation case. Relative differences increase commensurately with the absolute differences towards the end of the 2 hour simulation and reach as high as 40% for the 3D thermal radiation case.

We then separated the rain rates according to different factors that could affect droplet growth on our trajectories. Following on the results of our investigation of the individual trajectories, we calculated rain rates for parcels with certain thresholds of updraft speeds, cumulative cooling, or time spent at cloud side. About 50% of the rain rate arises from parcels that are in an updraft region of 3 ms$^{-1}$ or more (regions typically associated with higher $q_c$), but differences between the NR and 1DR/3DR

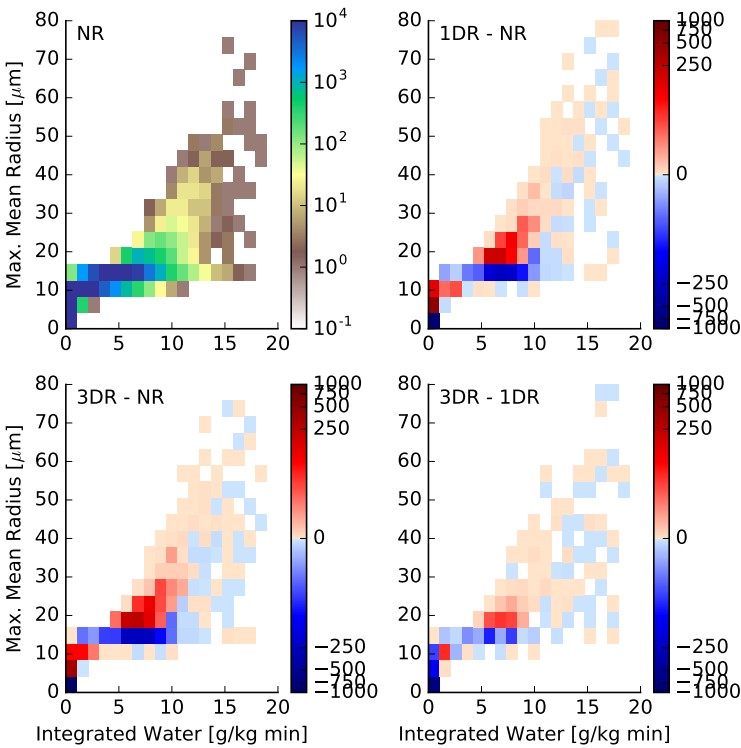

**Figure 11.** Joint histogram of integrated water and maximum mean radius. The first figure shows all data from the NR simulation. The other figures show the difference of the number of occurrences of 1DR vs NR, 3DR vs. NR and 3DR vs. 1DR.

cases are small. The largest radiative effect emerges from parcels that recirculate (see Fig. 14). We define 'recirculation' by setting a threshold in terms of $q_c$ of 0.01 gkg$^{-1}$.

The time periods for recirculation events are shown in Fig. 13. Most of the parcels spend a few minutes outside a cloud. More than 90% of the recirculation events are shorter than 5 minutes. Up to 58% of the parcel rain rate of the 3DR simulation arises

5  from recirculating parcels, while in the case of NR, about 45% of the parcel rain rate arises from recirculating parcels. The largest increase is found within the last 20 minutes of the investigated timeframe. Differences of 5-10% are found in the first 20 to 50 minutes, while in between, there is no difference between the three simulation types. Setting an upper limit in time (e.g. 5 min, which includes more than 90% of our recirculating parcels), the changes in our results are very small. The maximum contribution of the rain rate reduces to 56%. For a time threshold of 2 min, the maximum reduces to 45%. In this context it

10  should be noted that only 6-7% of our 340,000 parcel trajectories are classified as recirculating according to our definition. Remarkably, these 6-7% can contribute up to 60% of the total parcel rain rate. The parcel rain rate due to recirculation (when normalized to each of the corresponding simulations and therefore without considering radiative effects) is about 30% in our study. Naumann and Seifert (2016) found a similar magnitude in their study. About 50% of the rain rate emerged from recirculating parcels. These zones might be considered the birth place of precipitation embryos, which, subsequently become

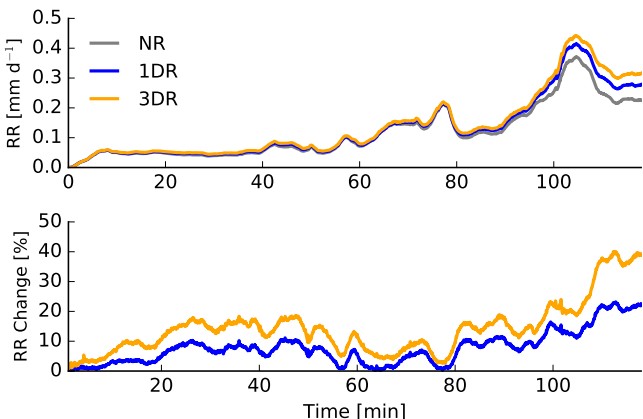

**Figure 12.** Total parcel rain rate calculated from the trajectory ensemble simulation (accounting for drop radii $> 20 \ \mu$m ). The top panel shows the absolute rain rate for the NR, the 1DR and the 3DR cases. The bottom panel shows the relative differences of 1DR simulation and the NR case, and 3DR simulation and the NR cases.

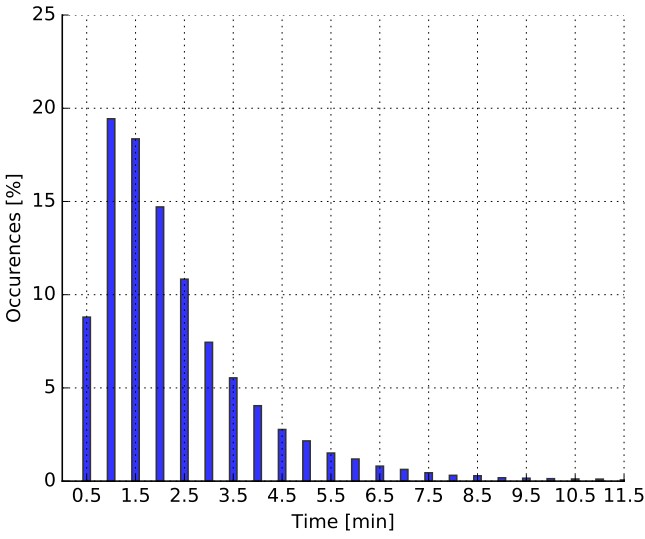

**Figure 13.** Histogram of the time period of recirculation events. Recirculation events are defined by threshold in terms of $q_c$ of 0.01 gkg$^{-1}$.

important for accelerating collision and coalescence.

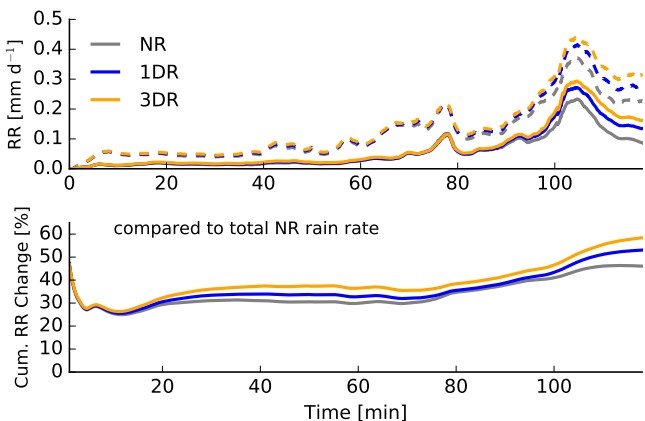

**Figure 14.** Total rain rate calculated from the trajectory ensemble simulation from recirculating parcels with a threshold of $0.01$ gkg$^{-1}$. The dashed lines show the total rain rate, the solid line the rain rate from recirculating parcels as well as the relative difference between the cumulated sum rain rates from recirculating parcels compared to the cumulated sum total NR rain rate.

## 4.3 Coupled LES Simulation - Cloud Droplet Growth under the Impact of Thermal Radiative Effects

Next, we investigate the effect of 1D and 3D thermal radiation in a fully coupled system. To this end, we ran a set of BOMEX simulations as described in Section 3. Here we a) look at the effects of thermal radiation on microphysics in a coupled system and b) compare it to the effect on dynamics.

### 4.3.1 The Effect of Thermal Radiative Transfer on Microphysics

We start with variables concerning rain, and first focus on the microphysical effect. Figure 15 shows rain water path, surface precipitation fraction, domain-averaged surface precipitation rate and the cumulative surface precipitation rate of 4 of the 5 simulations. We take the simulation with 1D radiation on dynamics ($1DD$) as our reference case. We focus first on the discus-
10  sion of the rain water path. We find a small increase in rain water path an hour after restart in the case of 3D radiation acting on microphysics and dynamics ($3DD\_3DM$), however differences among the four simulations never become significant. Surface precipitation fraction (over the total domain) shows an increase for all simulations where radiation is coupled to the diffusional droplet growth. The strongest increase is found for the $3DD\_3DM$ case. This suggests that more clouds produce rain when radiation is coupled to the droplet growth, but the total amount of rain water produced does not change substantially.
15  The surface precipitation rate shows no clear changes, but in accumulation, the simulations with the radiative-microphysical coupling produce more rain ( $10\%$ for $3DD\_3DM$). There is a subtle increase in the accumulated rain rate and rain water path when thermal radiation is coupled to the diffusional droplet growth. However, when comparing $1DD\_3DM$ to $1DD\_1DM$, no difference can be found. Hence, the small increase in rain in the case of $3DD\_3DM$ must arise from the effects of 3D

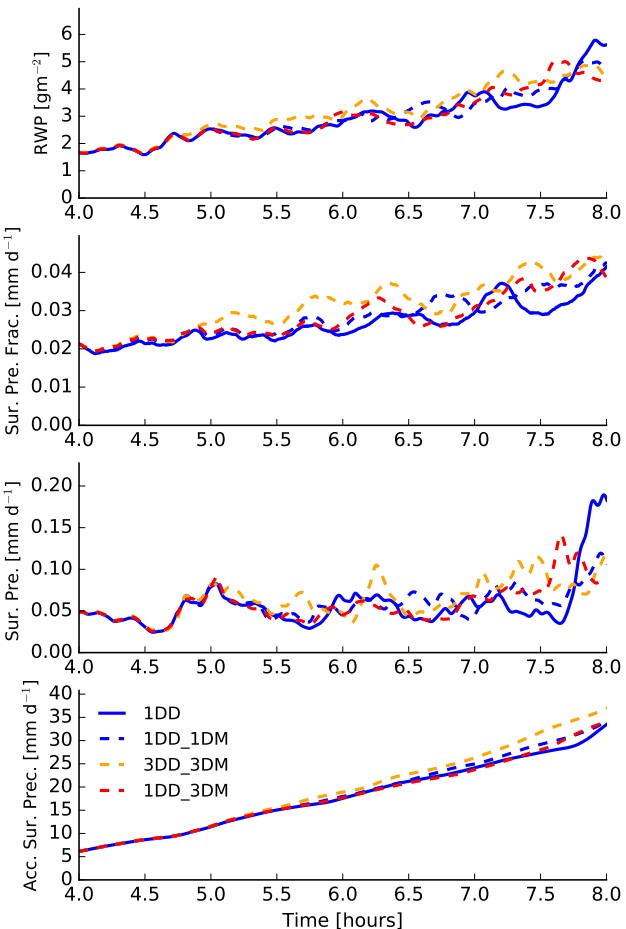

**Figure 15.** Temporal development of rain water path (top), surface precipitation fraction (second from top), domain averaged surface precipitation rate (second from bottom) and the accumulated domain averaged surface precipitation rate (bottom). The time series is shown from the restart time of 4 hours onward. We compare four of our five simulations here where thermal radiation was coupled to the diffusional droplet growth.

thermal radiation on dynamics, not on microphysics. We will investigate the 3D effect further in the following.

### 4.3.2    3D Thermal Radiative Effects

Figure 16 shows the same variables as Fig. 15 but now comparing the results of the $3DD\_3DM$ and $3DD$. In the beginning, rain water path and rain rate show no noticeable difference. After 7 hours of the simulation, rain water increases for $3DD$. Prior to 7 hours, as also discussed in 4.3.1, the fraction of surface rain rate is slightly enhanced in $3DD\_3DM$ compared to $3DD$, which again suggests that the coupling to microphysics does not produce more rain but that more clouds produce small amounts of rain, while in the $3DD$ case changes in the dynamics cause somewhat stronger rain in fewer clouds. The lower figure shows again the accumulated surface rain rate over time. The $3DD$ simulation produces more rain. This was counterintuitive at first, because it was expected that thermal radiative effects enhance droplet growth.

We pose the following hypothesis to explain this behavior:

*Enhanced droplet growth, due to 3D thermal radiation at the cloud edges decreases the evaporation at the cloud edges causing weaker evaporative cooling, and weaker downward motion, representing an evaporation-circulation feedback*

This hypothesis constitutes a negative feedback to changes in the cloud circulation found by Klinger et al. (2017), and will be explained in the following. It is analogous to the previously documented evaporation-circulation feedback due to changing aerosol concentrations in cumulus clouds (Xue and Feingold, 2006) and earlier studies that identified the relationship between the horizontal buoyancy gradient and the vortical circulation around a cloud; stronger cloud edge evaporation generates stronger horizontal buoyancy gradients, increased TKE and enhanced mixing and entrainment (Zhao and Austin, 2005).

Klinger et al. (2017) found an enhanced cloud circulation due to thermal radiative effects. It was shown that cloud top cooling caused stronger updraft velocities in the clouds, and due to the side cooling stronger subsiding shells at the cloud edge. Due to the stronger updrafts, clouds were deeper, more turbulent, and contained more $q_c$. The results from Klinger et al. (2017) and the above posed hypothesis, can, if correct, explain the differences in surface rain fraction and rain rate between the two simulations. We will therefore investigate the profiles of cloud water mixing ratio, precipitation flux, evaporation rate and buoyancy production of TKE (Fig. 17), averaged over half an hour marked by the gray shading in Fig. 16. This time period is chosen as it is the period shortly before and at the beginning of the increase in rain production. All four variables show higher values for $3DD$ compared to $3DD\_3DM$. Finally, we look at the temporally averaged profiles of updraft and downdraft vertical velocities in saturated areas. Here we also find stronger downdraft and stronger updrafts in the $3DD$ case. These analyses lend credence to our hypothesis, and Klinger et al. (2017) (Fig. 18).

### 4.3.3    1D vs 3D Thermal Radiative Effects

Finally, we compare the results of $1DD$ and $3DD$ to examine the effect of 3D thermal radiation on dynamics. We focus again on rain production as this was not included in Klinger et al. (2017). As expected, 3D radiation causes an increase in all the rain-related variables (shown in Fig. 19). To prove that a change in the cloud circulation is also causing the increase in rain, we look at the profiles of the updraft and downdraft vertical velocity in saturated areas, averaged over two time periods marked in

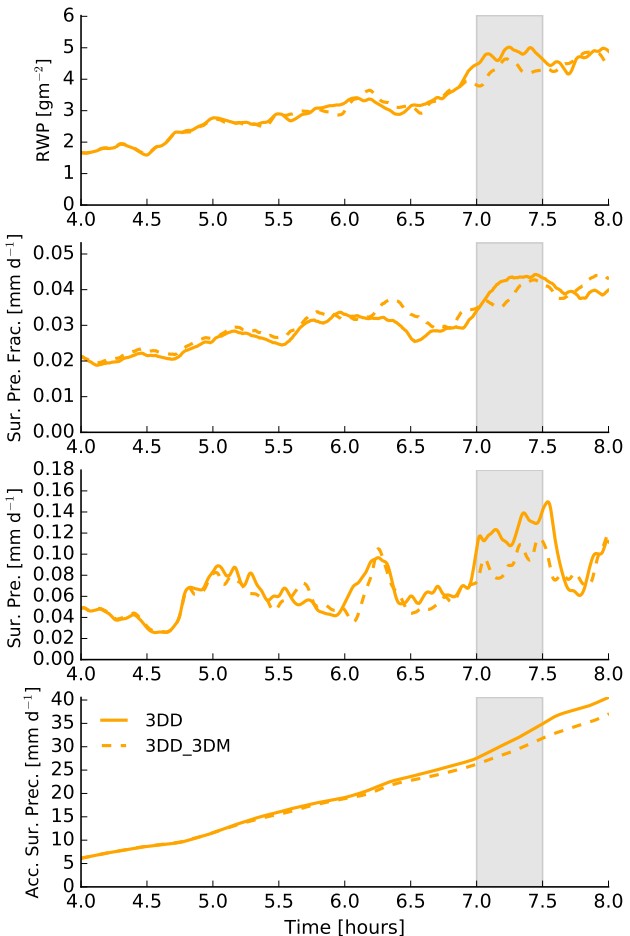

**Figure 16.** Similar to Figure 15 but for $3DD$ and $3DD\_3DM$. The gray shaded area show the time period between 6.7 and 7.3 hours which is investigated in the further analysis.

gray in Fig. 19. The time periods were again chosen because they include the beginning of rain enhancement by 3D thermal radiation compared to 1D thermal radiation. For both time periods, Fig. 20 shows enhanced downdrafts and updrafts for the $3DD$ simulation compared to $1DD$.

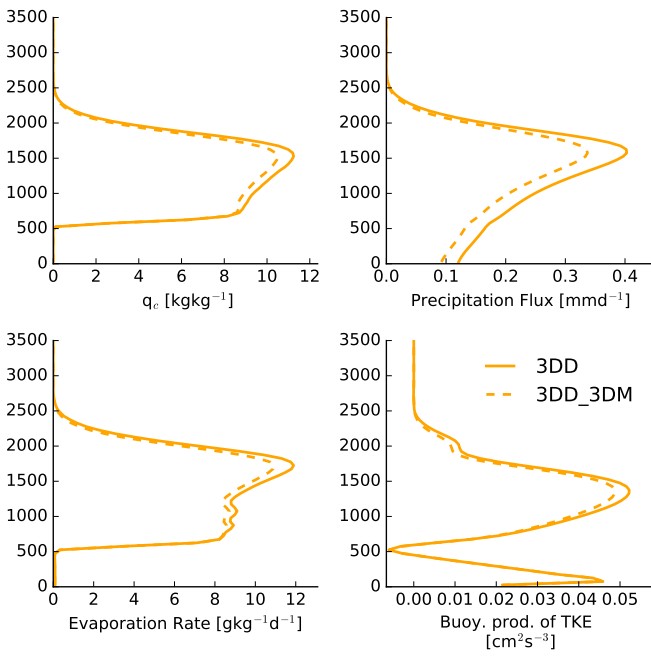

**Figure 17.** Time averaged profiles of cloud water mixing ratio, precipitation flux, evaporation rate and buoyancy production of the TKE averaged from 6.7 to 7.3 hours.

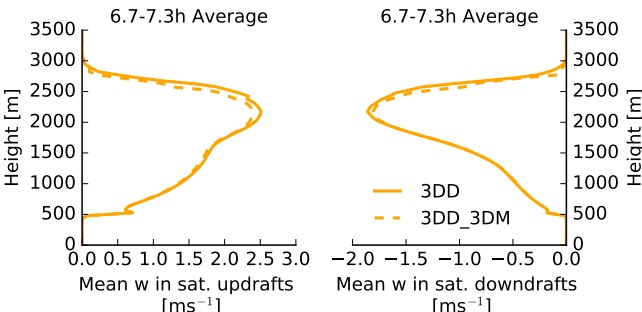

**Figure 18.** Time averaged profiles of updraft and downdraft vertical velocity in saturated areas averaged from 6.7 to 7.3 hours.

### 4.3.4 Summary

The coupling of thermal radiative effects to microphysics can lead to the formation of larger cloud droplets and drizzle droplets. For 3D thermal radiative effects, additioinal cooling occurs at cloud edges which can strengthen the effect. The coupled LES simulations showed that:

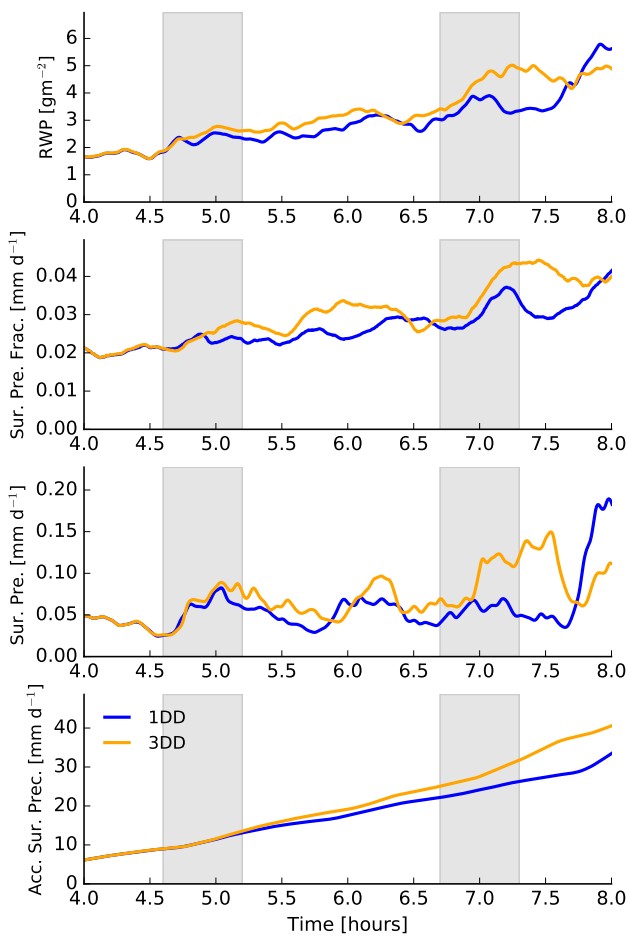

**Figure 19.** Similar to Fig. 15 but for $1DD$ and $3DD$. The gray shaded area show the time period between 4.6 and 5.3 hours as well as 6.7 and 7.3 hours which are investigated in the further analysis.

– When thermal radiation is coupled to microphysics, there is a small increase in rain production for 1D radiative effects ($1DD\_1DM$). This could be due to recirculation of droplets, as shown in the parcel model study. However, the change of rain in the coupled simulations is very small. When coupling radiation to droplet growth it matters little whether 3D or 1D thermal radiation is applied. The increase in the surface precipitation fraction suggests that rain is produced in more clouds distributed over the domain.

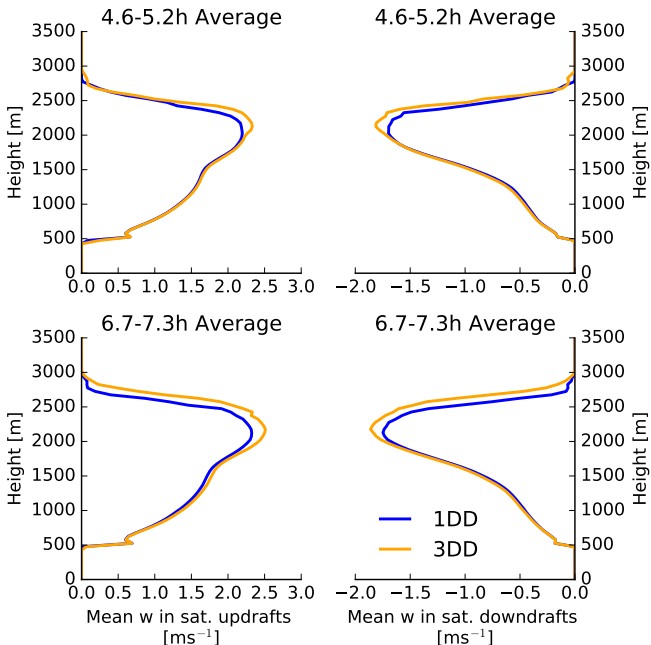

**Figure 20.** Similar to Fig. 18 but for $1DD$ and $3DD$ and for the time period between 4.6 and 5.3 hours as well as 6.7 and 7.3 hours.

- When 3D thermal radiative effects are considered we find (counterintuitively) overall more rain in the simulation with dynamics only.

- The fact that more rain is produced by the simulation coupled to dynamics only is hypothesized to be due to an evaporation-circulation feedback caused by the larger droplets in the $3DD\_3DM$ simulation.

- When comparing the 1D and 3D thermal radiative effects on dynamics we find an increase in the rain production for 3D thermal radiation.

- The dynamical effect caused by 1D and 3D thermal radiation is a change in the cloud circulation as already found by Klinger et al. (2017) where thermal radiation increases upward and downward vertical velocities in and in the near cloud environment, which causes deepening and more rain.

10 Finally, we note that the overall differences concerning precipitation are small and might not be detectable relative to differences associated with perturbations to thermodynamical inputs in an ensemble of simulations (see eg. Lonitz et al. (2015)).

## 5 Conclusions

In this study we investigated the effect of thermal radiation on cloud droplet growth and rain formation in shallow cumulus clouds. We used a two stage approach, which allowed us to separate microphysical from dynamical effects.

First an offline parcel model was used to investigate the effect of 1D and 3D thermal radiation on cloud droplet growth. It was found that thermal radiation in general has the potential to enhance droplet growth and rain formation. 3D thermal radiation enhances droplet growth and rain formation more than 1D thermal radiation. It was shown that thermal radiation enhances the formation of precipitation embryos in the 10-30 $\mu$m radius range. These embryos have the potential to enhance rain formation

in real clouds. Thermal radiation can affect cloud droplet growth when one or more of the following conditions are fulfilled: Droplets have already grown to a size of about 10 $\mu$m or more when being exposed to thermal cooling. A cooling period of more than 5 minutes at a cooling rate of 20 Kd$^{-1}$ or more, and vertical velocities close to zero are favorable for radiative effects. If one or more of these factors occur, radiative cooling can enhance droplet growth. The main effect was found in re-circulating parcels, which fulfill parts of the above-mentioned criteria. Recirculating parcels include droplets that have already

grown to a certain size when passing a cloud edge area. At the cloud edge area cloud droplets are exposed to large cooling (close to black body emission) and this small number of droplets grows by thermal cooling, which counteracts evaporation. When reentering a cloud, droplet growth continues, generating a broader spectrum, which can enhance rain formation. Only 6-7% of our simulated parcels are classified as recirculating, yet they can contribute up to 45-60% of the local rain rate.

Second, in a more realistic framework we investigated large eddy simulations where thermal radiative effects were applied to

15 droplet growth and dynamics. It was shown that the effect on droplet growth is small. However more clouds produce small amounts of rain when radiative effects are applied to the diffusional droplet growth; thus rain covers a larger area of the simulation domain. 3D thermal radiative effects exceed 1D thermal radiative effects. The largest amount of rain is produced when 3D thermal radiation is applied to dynamics only. This was initially considered to be counterintuitive since both microphysical and radiative effects tend to enhance rain. We hypothesize that an evaporation-circulation feedback is responsible for less rain

in the simulation where radiation is also applied to droplet growth: 3D thermal cooling rates at the cloud edges enhance droplet growth locally at the cloud edge, thus leading to weaker evaporation rates, which in turn reduces the strength of the subsiding shell and the horizontal buoyancy gradient, all leading to weaker cloud turbulence and lower rain production. In simulations with 3D thermal cooling only applied to the dynamics, the enhanced cloud circulation causes stronger updrafts in the cloud center, a cloud deepening and more condensation/ rain formation.

These results could have implications in terms of cloud field organization. As shown by Klinger et al. (2017) thermal radiation can cause mesoscale organization of shallow cumulus clouds by changing cloud circulation. It was shown that this change in cloud circulation also occurs in the simulation in this study. Furthermore, more rain produced by thermal radiation changes the dynamics of the system as a whole. The larger area of the domain covered by rain when radiative effects are applied to micro-physics could also lead to a feedback in terms of dynamics and cloud field organization. Longer simulations are necessary to

investigate the organization feedbacks. Finally, a trade-wind cumulus case that tends to deepen and generate more precipitation and organization would be worth investigating.

## Appendix A: Timescale Calculation

The analysis of timescales supports our argument about the usefulness of the the parcel model, even though it does not represent certain physical processes. The characteristic timescales for a number of processes involved in our study (diffusional droplet growth, diffusional droplet growth with radiation, and sedimentation) are calculated in the following. The analysis is performed for a droplet of 20 $\mu m$ radius. In order to avoid confusion, we use $\chi$ to represent timescales, because the standard abbreviation $\tau$ is already used as the forcing in the diffusional droplet growth equation (Eq.7).

The analysis shows a clear signal: sedimentation occurs on much longer timescales than all the other processes. The timescale of sedimentation is on the order of 1 hour, while the diffusional droplet growth timescales, with and without radiation, are on the order of a few minutes.

### Diffusional Droplet Growth

Diffusional droplet growth (which follows from Eq. 1) is definded as

$$\frac{dr}{dt} = \frac{1}{\rho C} \cdot \frac{S}{r}. \tag{A1}$$

It therefore follows that the charcteristic timescale $\chi_{growth}$ is

$$\chi_{growth} = \frac{r^2 \cdot C}{S} = 6 \; min \; 40 \; s \tag{A2}$$

with a droplet radius $r$ of 20 $\mu m$, $C = 1e^{10} \; sm^{-2}$ and a supersaturation $S = 0.01$.

### Diffusional Droplet Growth including Thermal Radiative Effects

The diffusional droplet growth including radiative effects (see also Eq. 2, but defined in mass space) is

$$\frac{dr}{dt} = \frac{1}{\rho C}(\frac{S}{r} - F \cdot E_{net}). \tag{A3}$$

Therefore $\chi_{growth,rad}$ becomes (when assuming that $r$ is constant over time in the radiative term)

$$\chi_{growth,rad} = \frac{r^2 \cdot C}{S - r \cdot F \cdot E_{net}} = 5 \; min \; 30 \; s \tag{A4}$$

with $F \approx 1$ and $E_{net} = \sigma \, T^4 \cdot 0.33 \approx -100 \; Wm^{-2}$ (on third of the black body radiation occuring in the window region).

### Sedimentation

The timescale for sedimentation follows

$$\chi_{sed} = \frac{L}{v} = 1 \; h \; 23 \; min \tag{A5}$$

with a typical lenght scale of $L = 100m$ and a fall velocity at $20\mu m$ of $v(20 \; \mu m) = 0.02 \; ms^{-1}$.

*Code availability.* Input files and the model code for reproducing the simulations and data of this study are available from the corresponding author upon request.

*Competing interests.* The authors declare that they have no conflict of interest.

*Author contributions.* CK implemented the 3D radiative transfer scheme into the LES, ran the simulations and performed the analysis. TY
5  implemented the bin microphysics scheme into the LES. All authors contributed to developing the basic ideas, discussing the results, and preparing the manuscript.

*Acknowledgements.* The authors acknowledge Bernhard Mayer and Jerry Harrington for useful discussion and Jan Kazil for help with SAM. We gratefully acknowledge Marat Khairoutdinov (Stony Brook University) for developing and making the System for Atmospheric Modeling (SAM) available. The project was founded by the 'German Research Foundation' (DFG, Research Fellowship KL-3035/1) and the Federal
10  Ministry of Education and Research (BMBF) through the High Definition Clouds and Precipitation for Climate Prediction project (HD(CP)$^2$) phase 2 (Förderkennzeichen: 01LK1504D). The authors acknowledge the NOAA Research and Development High Performance Computing Program for providing computing and storage resources that have contributed to the research results of this paper.

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
