# Peer review of "Cloud Droplet Growth in Shallow Cumulus Clouds Considering 1D and 3D Thermal Radiative Effects"

_Atmospheric Chemistry and Physics, 2018_

## Referee Comment (RC1) · Anonymous Referee #1 · 20 Dec 2018

This study explores the effect of radiation on cloud droplet growth in shallow cumulus clouds. Using an offline parcel model it is found that radiative cooling affects droplet growth mostly for drop sizes larger than 10 $\mu$m and that the effect is stronger for 3D than for 1D radiation. Recirculating parcel meet several of the criteria that are found to be favorable for a strong effect of radiative cooling on cloud droplet growth and hence contribute more strongly to rain production than if radiative effects on diffusional droplet growth is not taken into account. In a sencond part, a fully coupled LES simulation with bin microphysics is used to unravel effects of 1D and 3D radiation on the dynamics and the diffusional growth. Although the differences in rain production are overall small, the authors conclude that radiative effects on microphysics increase the rain fraction but that a dynamic evaporation-circulation feedback can decrease rain amount.

[Figure]

The topic of the study is very interesting and relevant for ACP. The manuscript is overall written well and the plots illustrate their content well. However, I have two issue with the study in its present form: First, the authors themselves argue that because in the parcel model raindrops are not allowed to fall out of their parcel, the parcel model is not appropriate to analyze rain formation. Nevertheless parts of their analysis (Fig. 6, 7, 9, 12) focuses on drops of sizes that have considerable fall speed and on rain rate. Second, the differences in the coupled LES simulations are very small. Because simulated precipitation from shallow clouds is such a sensitive variable, I am sceptical that these small differences are robust and due to physical causes rather than random fluctuations due to different realisations. I comment on these two issues in more detail below followed by some specific and technical comments.

1) In the method section you argue (and I agree) that the methodology is only valid until the onset of drizzle. Analysing the rain rate (Fig. 12) is therefore not appropriate. In Fig. 6 and 9 the main differences are seen for drops larger than 20 $\mu$m radius, which is just the size where cloud droplets turn into drizzle (e.g., Sant et al., 2013, JAS). Even with a less conservative estimate drops with a radius of 40 $\mu$m develop considerable fall speed. In the parcel model, those larger drops do not fall out but instead keep on interacting with the smaller drops in their parcel, which is unphysical. Because of the interaction of the large drops with the small droplets, changes in the smaller part of the drop size spectrum are not independent of this issue as soon as some larger drops form.

2) In the coupled LES simulations overall differences are small (e.g., Fig. 14 and 15) and I am wondering whether they are actually causally related to the changes in the radiation. My null-hypothesis would be that the differences in the runs are due to random fluctuations in different realisations. One way to test this is to run an ensemble of simulations for each of the five modifications and analyze whether those ensembles (and their spread) are significantly different from each other.

Specific comments:

- While I found the rest of the paper well-written and clear, Section 2 was confusing to me. I am not a radiation expert and as other reader might not be either, some improvement is needed here. Please, help the reader by systematically arguing why you need each equation and what it is leading to. E.g., why do you need the forcing term tau and in which equation does it go? On p.5 l.11 you arrive at the same expression you already had on p.3 l.20. Which equation do you use in your model? Also, make sure you explain all variables and constants (see also my technical comment).

- In the first line of Eq 4, why is there no sum over the different sizes of m within the bin range of bin k? Under which assumptions does the approximation hold? Please use $\approx$ where necessary to indicate the approximation.

- p.6 l.2: Are the trajectories run as tracers that go with the flow or is an average fall velocity of particles size distribution taken into account? From what I read later, I assume they are calculated as tracers of the flow. This would be helpful to specify already here.

- How are the cloud droplet size distributions initialised in the offline parcel model?

- Is the analyzed parcel in Fig 4 and 5 the same? This is not clear from the text.

- Fig. 7, tau: It would be interesting to calculate the overall contribution to droplet growth for each of the three sizes from the dynamic and the radiative term. The values of the radiative forcing are much smaller than the dynamic terms but the dynamical forcing also has substantial negative contributions.

- p.13 l.8: This sentence is not clear to me: "The radiative cooling does not seem to cause droplet growth in individual parcels beyond the NR case..." Are you saying that in parcels where there are no cloud droplets for NR, also no cloud droplets will grow with 1DR or 3DR?

- Fig 12 and 13: Is this the rain rate at the surface or integrated over all heights? I assume the latter. Please clarify. However, given the restriction of the method to the

onset of drizzle, I do not think it is appropriate to analyze the rain rate (see my general comment).

- p.16 l.20: How long do the parcels need to be in an environment with qc less than 0.01 g/kg to be called a recirculation? Do your statistics substantially change if you increase or decrease a threshold in time a bit?

- To quantify how much recirculation parcels contribute, I think it makes more sense to normalize by the total rain rate for each simulation, e.g., RR_1DR_recirculating/RR_1DR_total. Otherwise you combine the effect of radiation and recirculation.

- p.18 l.9 and elsewhere: Please do not use "significant" when you did not test for significance. "considerable" or "substantial" might be alternatives.

- Fig. 14,15,18: Why don't you combine those three figures into one? If I am not mistaken you just need to add one line in Fig. 14. It would allow the reader to compare simulation pairs, which the manuscript does not focus on. Also, I do not see the advantage of comparing different time periods for the different simulations in Fig. 17 and 19. If the results are robust, I would expect the results to be qualitatively the same for other time periods. Then Fig. 17 and 19 could be combined and show all five simulations. Also, show all five simulations in Fig. 16.

- Section 4: If the results turn out to be robust (see my general comment), my interpretation here would be that applying radiative cooling to microphysics leads to larger cloud droplets and drizzle drops especially at the cloud edges, which then can lead to two opposing effect: 1) The larger drops re-enter the cloud and have a better chance to form rain than if they had not grown by radiation. Therefore rain formation is enhanced. 2) The larger drops decrease evaporation at the cloud edge, which via the evaporation-circulation feedback leads to weaker updrafts and hence less rain formation. Going from 1DD to 1DD_1DM effect 1 seems to dominate; going from 3DD to 3DD_3DM effect 2 seems to dominate. If you agree with this interpretation, I think it

would be worth to make this opposing effects clearer in the manuscript. Here, it would strengthen your point if you analyze the evaporation rates for all simulations (Fig. 16). Also, I suggest to contrast the difference in evaporative cooling at the cloud edge with the difference in radiative cooling rates at cloud edge to unravel whether a decrease in evaporative cooling or the increase in radiative cooling dominates if you go from 1DD to 3DD_3DM.

- p.1 l.11 "Small amounts of rain are ..." and p. 24 l.12 "... rain covers a larger area": I do not see that these sentences are true for 3DD to 3DD_3DM (Fig. 14).

Technical comments:

- Please check the standards of the journal. Usually references should be ordered by year (e.g., p.3 l.6 or l.10, check elsewhere) and brackets in brackets are not allowed.

- pay attention if you place variables (math mode) in the text, e.g., $T_\mathrm{inf}$ on p.3 l.17 or $q$ on l.27

- make sure to explain all variables and constants, e.g., D and e_s on p.3 l.28 or t_f on p.4 l.13

- p.6 l.7: Harrington et al. (2000)

- caption Fig.4: where there parcel stays -> where the parcel stays

- Fig 4: What the color scale of qc in b-d? It seems not the be the same as for the trajectory in a.

- p.11 l.4: four -> three

- p.12 l.12: the the -> the

- p.18 l.24: stronger stronger -> stronger

---

## Referee Comment (RC2) · Anonymous Referee #2 · 30 Dec 2018

I support publication. The manuscript is quite well written as it is. There are a few relatively minor points that the authors may wish to consider that I have raised below, but, overall, the manuscript is in good shape.

It seems the question as to why cloud droplet spectra are "broad" has been plaguing the cloud physics community since we were first able to compare measured droplet distributions with calculations of adiabatic growth. There are many competing hypotheses as to why this discrepancy exists, which the authors have referenced in their introduction. In my opinion, one reason that we may have been puzzled about this for so long is that there's no single reason for this spectral broadening, and in most cases, it is probably a combination of factors. Radiative effects certainly play some role, and this paper shows some of the effects for shallow cumulus clouds.

**Misc. comments**

pg. 1, last line: "Smaller droplet sizes reflect more radiation back to space..." Clarify please. Perhaps just add that this is true if the liquid water is held constant.

References to Pruppacher and Klett: This is a 700 page book. At least reference the chapter for the point you are trying to make.

Pg. 2; line 1: "...larger droplets allow radiation to settle more easily..." Settle implies that gravity is the driving force. I think that you mean that radiation penetrates to the surface more easily when the droplets are larger.

Pg. 3, line 5: You reference high local cooling rates for cumulus clouds in the context of the finite size. It would be useful to provide a value for typical cooling rates for stratus clouds at cloud top.

Pg. 6, line 15: Check the parentheses on the reference to Harrington.

Pg. 7, line 6: The first two sentences are redundant. One or the other is fine.

Pg. 12, line 12: Two consecutive occurrences of "the".

Pg. 18, line 24: Two consecutive occurrences of "stronger".

Pg. 17 (last line) continuing to first lines of pg. 18: In the discussion of Figure 14, the statement is made that the rain water path shows an increase an hour after restart for the 3DD_3DM case. Is that a significant difference? I can see that the line is slightly higher than the other three, but the difference disappears after time 6.5. In the text, that's attributed to noise, but then why isn't the initial increase just noise as well?

---

## Author Comment (AC1) · 6 Mar 2019

We thank reviewer 1 for the helpful comments which helped us to improve the manuscript.

*This study explores the effect of radiation on cloud droplet growth in shallow cumulus clouds. Using an offline parcel model it is found that radiative cooling affects droplet growth mostly for drop sizes larger than 10 $\mu$m and that the effect is stronger for 3D than for 1D radiation. Recirculating parcel meet several of the criteria that are found to be favorable for a strong effect of radiative cooling on cloud droplet growth and hence contribute more strongly to rain production than if radiative effects on diffusional droplet growth is not taken into account. In a second part, a fully coupled LES simulation with bin microphysics is used to unravel effects of 1D and 3D radiation on the dynamics and the diffusional growth. Although the differences in rain production are overall small, the authors conclude that radiative effects on microphysics increase the rain fraction but that a dynamic evaporation-circulation feedback can decrease rain amount.*
*The topic of the study is very interesting and relevant for ACP. The manuscript is overall written well and the plots illustrate their content well. However, I have two issue with the study in its present form: First, the authors themselves argue that because in the parcel model raindrops are not allowed to fall out of their parcel, the parcel model is not appropriate to analyze rain formation. Nevertheless parts of their analysis (Fig. 6, 7, 9, 12) focuses on drops of sizes that have considerable fall speed and on rain rate. Second, the differences in the coupled LES simulations are very small. Because simulated precipitation from shallow clouds is such a sensitive variable, I am sceptical that these small differences are robust and due to physical causes rather than random fluctuations due to different realisations. I comment on these two issues in more detail below followed by some specific and technical comments.*
* * *
*1) In the method section you argue (and I agree) that the methodology is only valid until the onset of drizzle. Analysing the rain rate (Fig. 12) is therefore not appropriate. In Fig. 6 and 9 the main differences are seen for drops larger than 20 $\mu$m radius, which is just the size where cloud droplets turn into drizzle (e.g., Sant et al., 2013, JAS). Even with a less conservative estimate drops with a radius of 40 $\mu$m develop considerable fall speed. In the parcel model, those larger drops do not fall out but instead keep on interacting with the smaller drops in their parcel, which is unphysical. Because of the interaction of the large drops with the small droplets, changes in the smaller part of the drop size spectrum are not independent of this issue as soon as some larger drops form.*
* * *
We agree that the parcel model is an idealized framework where some processes, as e.g. the precipitation formation are not treated in the most realistic way. However, it is a suitable framework for the detailed examination of certain processes. It offers the possibility for a detailed investigation of the differences related to radiation in this study which would be impossible to investigate in an LES. For a more realistic representation of the onset of precip we use the LES in part two of this study.

*2) In the coupled LES simulations overall differences are small (e.g., Fig. 14 and 15) and I am wondering whether they are actually causally related to the changes in the radiation. My null-hypothesis would be that the differences in the runs are due to random fluctuations in different realisations. One way to test this is to run an ensemble of simulations for each of the five modifications and analyze whether those ensembles (and their spread) are significantly different from each other.*
* * *
We agree with the reviewer that the overall differences are small and might not be detectable relative to differences associated with perturbations to thermodynamical inputs in an ensemble of simulations (e.g. Lonitz et al., JAS, 2015). We now state this clearly in the revised manuscript. An ensemble of 3 remembers for the 5 LES runs would require an additional 10 simulations, which would be computationally unfeasible.

***Specific comments:***

*- While I found the rest of the paper well-written and clear, Section 2 was confusing to me. I am not a radiation expert and as other reader might not be either, some improvement is needed here. Please, help the reader by systematically arguing why you need each equation and what it is leading to. E.g., why do you need the forcing term tau and in which equation does it go? On p.5 l.11 you arrive at the same expression you already had on p.3 l.20. Which equation do you use in your model? Also, make sure you explain all variables and constants (see also my technical comment).*
* * *
Up to equation 6, we essentially follow Harrington et al., 2000. Therefore we summarize only briefly the most important steps/equations. We added some further description and point directly to Harrington et al., 2000 and the necessary equations therein.
From Equation 7 on we describe how the radiative term that is necessary in the droplet growth equation can be derived correctly from the LES model heating rate. We also added further information on why we need these steps and trust that the chapter is more understandable now.

*- In the first line of Eq 4, why is there no sum over the different sizes of m within the bin range of bin k? Under which assumptions does the approximation hold? Please use $\approx$ where necessary to indicate the approximation.*
* * *
This is because for each bin, the equation is solved for the mean mass of the bin, a quantity available in a 2-moment bin scheme but not in typical one moment bin schemes.

*- p.6 l.2: Are the trajectories run as tracers that follow the flow or is an average fall velocity of particles size distribution taken into account? From what I read later, I assume they are calculated as tracers of the flow. This would be helpful to specify already here.*
* * *
The trajectories are run as tracers that go with the flow. We corrected the sentence to: "For the first part of the study 2.7 million Lagrangian air parcel trajectories were recorded in the last two hours of the BOMEX simulation with a 2~second time step."

*- How are the cloud droplet size distributions initialized in the offline parcel model?*
* * *
We use a log-normal distribution. We added this to the manuscript.

*- Is the analyzed parcel in Fig 4 and 5 the same? This is not clear from the text.*
* * *
Yes, both figures show the same parcel. We changed the beginning of the paragraph to the following to be more precise:

"We now take a more detailed look at the same parcel trajectory as shown in (Fig.4). Figure 5 shows this selected parcel which is characterized by moderate vertical velocities..."

*- Fig. 7, tau: It would be interesting to calculate the overall contribution to droplet growth for each of the three sizes from the dynamic and the radiative term. The values of the radiative forcing are much smaller than the dynamic terms but the dynamical forcing also has substantial negative contributions.*
* * *
The overall contribution of the radiative term is positive, while the dynamical forcing undergoes positive and negative terms along this parcel trajectory. Both forcings, however act locally in time as shown in Figure 7.

*- p.13 l.8: This sentence is not clear to me: "The radiative cooling does not seem to cause droplet growth in individual parcels beyond the NR case..." Are you saying that in parcels where there are no cloud droplets for NR, also no cloud droplets will grow with 1DR or 3DR?*
* * *
Yes, this is what we meant. Radiation, acting on droplets itself does not cause additional droplet growth in cases where there was no rain, but strengthens existing droplet growth. However, in a dynamical sense, radiation can cause more rain formation (which is shown later on in the paper).

*- Fig 12 and 13: Is this the rain rate at the surface or integrated over all heights? I assume the latter. Please clarify. However, given the restriction of the method to the onset of drizzle, I do not think it is appropriate to analyze the rain rate (see my general comment).*
* * *
It is vertically integrated. We added this in the manuscript.

*- p.16 l.20: How long do the parcels need to be in an environment with qc less than 0.01 g/kg to be called a recirculation? Do your statistics substantially change if you increase or decrease a threshold in time a bit?*
* * *
We did not apply a threshold in time for the analysis, only for qc.

However, we did perform a "back of the envelope" calculation. Setting upper and lower thresholds in space (e.g. 50m or 200m for the "near cloud" environment) and speed (0.5m/s to 5 m/s), a time period for a parcel (if it travels straight back and forth) would be between 20s and 13min. Our parcels are within this range (see histogram at the end of this text, which we also added to the manuscript, now Figure 13).

Setting an upper limit in time (e.g. 5min, which is more than 90% of our recirculating parcels ), the changes in our results are very small. The maximum contribution of the rain rate 58% reduces to 56%. For a time threshold of 2min, the maximum reduces to 45%. We

added this analysis to the manuscript, too. For comparison, the second figure (similar to Fig. 14 in the paper) at the end of this text shows the rain rate for the 2min threshold.

*- To quantify how much recirculation parcels contribute, I think it makes*
*more sense to normalize by the total rain rate for each simulation, e.g.,*
*RR_1DR_recirculating/RR_1DR_total. Otherwise you combine the effect of radiation*
*and recirculation.*
* * *
This is a valuable comment. However, as our study is not purely focused on the effect of recirculation, but also includes the radiative effects, we prefer to show the figure as is. The change due to recirculation only is very much the same for the three cases (NR, 1D RAD and 3D RAD) and about 30%. We now mention this in the text as follows: "The amount of the rain rate due to recirculation (when normalized to each of our simulations and therefore without considering radiative effects) is about 30% in our study."

*- p.18 l.9 and elsewhere: Please do not use "significant" when you did not test for*
*significance. "considerable" or "substantial" might be alternatives.*
* * *
Corrected.

*- Fig. 14,15,18: Why don't you combine those three figures into one? If I am not mis-*
*taken you just need to add one line in Fig. 14. It would allow the reader to compare*
*simulation pairs, which the manuscript does not focus on. Also, I do not see the advan-*
*tage of comparing different time periods for the different simulations in Fig. 17 and 19.*
*If the results are robust, I would expect the results to be qualitatively the same for other*
*time periods. Then Fig. 17 and 19 could be combined and show all five simulations.*
*Also, show all five simulations in Fig. 16.*
* * *
Thank you for this comment. We do have reasons why we show the figure as they are, which are the following:
a) We want to point out different effects. This is why we show Figure 14, 15 and 18 the way they are. First, we only focus on thermal radiative effect on the droplet growth itself, not focusing on dynamical feedbacks (Figure 14 and Section 4.3.1). Second, we want to show the effect of 3D thermal radiation on dynamics and on microphysics (Figure 15 and Section 4.3.2). Finally, we separate 1D and 3D thermal radiative effects (Figure 18 and Section 4.3.3).
b) The different time periods are chosen, because they focus on specific/detailed developments within the simulations. In the case where we compare 3D radiative effects on microphysics and dynamics, an increase in rain arises roughly around 7 hours, therefore we expect changes in the variables related to the specific dynamics at this time (which is shown). The same reason applies to Figure 19. When comparing 1D and 3D thermal radiation, differences in both simulation occur at different time periods.

*- Section 4: If the results turn out to be robust (see my general comment), my inter-*
*pretation here would be that applying radiative cooling to microphysics leads to larger*
*cloud droplets and drizzle drops especially at the cloud edges, which then can lead to*
*two opposing effect: 1) The larger drops re-enter the cloud and have a better chance*
*to form rain than if they had not grown by radiation. Therefore rain formation is en-*

*hanced. 2) The larger drops decrease evaporation at the cloud edge, which via the evaporation-circulation feedback leads to weaker updrafts and hence less rain formation. Going from 1DD to 1DD_1DM effect 1 seems to dominate; going from 3DD to 3DD_3DM effect 2 seems to dominate. If you agree with this interpretation, I think it would be worth to make this opposing effects clearer in the manuscript. Here, it would strengthen your point if you analyze the evaporation rates for all simulations (Fig. 16). Also, I suggest to contrast the difference in evaporative cooling at the cloud edge with the difference in radiative cooling rates at cloud edge to unravel whether a decrease in evaporative cooling or the increase in radiative cooling dominates if you go from 1DD to 3DD_3DM.*
* * *
Thank you for summarizing this. We revisited the summary section and made these points clearer.

*- p.1 l.11 "Small amounts of rain are ..." and p. 24 l.12 "... rain covers a larger area": I do not see that these sentences are true for 3DD to 3DD_3DM (Fig. 14).*
* * *
When looking at the surface precipitation fraction (Fig. 15, second from top) the area covered by rain is slightly larger in the 3DD_3DM case compared to the 3DD case. The differences are small though, but visible throughout the simulation time until the dynamical effects dominate.

***Technical comments:***

*- Please check the standards of the journal. Usually references should be ordered by year (e.g., p.3 l.6 or l.10, check elsewhere) and brackets in brackets are not allowed.*
* * *
Corrected.

*- pay attention if you place variables (math mode) in the text, e.g., $T_\mathrm{inf}$ on p.3 l.17 or $q$ on l.27*
* * *
Corrected.

*- make sure to explain all variables and constants, e.g., D and e_s on p.3 l.28 or t_f on p.4 l.13*
* * *
Corrected.

*- p.6 l.7: Harrington et al. (2000)*
* * *
Corrected.

*- caption Fig.4: where there parcel stays -> where the parcel stays*
* * *
Corrected.

*- Fig 4: What the color scale of qc in b-d? It seems not the be the same as for the trajectory in a.*
* * *
Figure 4 b-c only shows the trajectory itself, not colored by qc or any other variable. We added this to the caption.

*- p.11 l.4: four -> three*
* * *
Corrected.

*- p.12 l.12: the the -> the*
* * *
Corrected.

*- p.18 l.24: stronger stronger -> stronger*
* * *
Corrected.

[Figure]

Histogram of recirculation times.

[Figure]

Rain rate calculated from the parcel ensemble (dashed) and from recirculating parcels (solid) applying a qc threshold of 0.01 g/kg and a time threshold for recirculation events of 2min.

---

## Author Comment (AC2) · 6 Mar 2019

We thank reviewer 2 for the helpful comments which helped us to improve the manuscript.

I support publication. The manuscript is quite well written as it is. There are a few relatively minor points that the authors may wish to consider that I have raised below, but, overall, the manuscript is in good shape.
It seems the question as to why cloud droplet spectra are "broad" has been plaguing the cloud physics community since we were first able to compare measured droplet distributions with calculations of adiabatic growth. There are many competing hypotheses as to why this discrepancy exists, which the authors have referenced in their introduction. In my opinion, one reason that we may have been puzzled about this for so long is that there's no single reason for this spectral broadening, and in most cases, it is probably a combination of factors. Radiative effects certainly play some role, and this paper shows some of the effects for shallow cumulus clouds.
* * *
*pg. 1, last line: "Smaller droplet sizes reflect more radiation back to space..." Clarify please. Perhaps just add that this is true if the liquid water is held constant.*
* * *
Thank you! We added: "for constant liquid water".

*References to Pruppacher and Klett: This is a 700 page book. At least reference the chapter for the point you are trying to make.*
* * *
Chapter 14, pages 569-616, is added to the citation.

*Pg. 2; line 1: "...larger droplets allow radiation to settle more easily..." Settle implies that gravity is the driving force. I think that you mean that radiation penetrates to the surface more easily when the droplets are larger.*
* * *
Thank you for pointing this out. We changed "settle" to "penetrate".

*Pg. 3, line 5: You reference high local cooling rates for cumulus clouds in the context of the finite size. It would be useful to provide a value for typical cooling rates for stratus clouds at cloud top.*
* * *
A comparison to stratus clouds in this context is difficult. Heating rates depend amongst other things on the temperature profile, liquid water content or, as they are volume quantities also on the grid box size of a simulation. But we see your point that some clarification for 1D and 3D radiative transfer cases is necessary. The already cited literature (and a new publication) compare heating rates in 3D and 1D in the same cloud fields based on the same background atmosphere and resolution.
We added: "Klinger and Mayer (2016, their Figure 11) showed that local peak differences in cooling rates between 1D and 3D thermal radiation in cumulus cloud fields can reach 20-120%, depending on the cloud field resolution. But the differences between 1D and 3D thermal radiation are not only focused on local grid boxes. Kablick et al. (2011) and Crnivec and Mayer (2019) showed that layer averaged 1D and 3D heating and cooling differences can be up to 1'K/d, which is the same order of magnitude as clear sky cooling."

*Pg. 6, line 15: Check the parentheses on the reference to Harrington.*
* * *
Checked.

*Pg. 7, line 6: The first two sentences are redundant. One or the other is fine.*
* * *
Corrected.

*Pg. 12, line 12: Two consecutive occurrences of "the".*
* * *
Corrected.

*Pg. 18, line 24: Two consecutive occurrences of "stronger".*
* * *
Corrected.

*Pg. 17 (last line) continuing to first lines of pg. 18: In the discussion of Figure 14, the statement is made that the rain water path shows an increase an hour after restart for the 3DD 3DM case. Is that a significant difference? I can see that the line is slightly higher than the other three, but the difference disappears after time 6.5. In the text, that's attributed to noise, but then why isn't the initial increase just noise as well?*
* * *
We agree that the word "noise" is misleading and have reworded the text accordingly.

---

## Author Response (AR2)

Dear Prof. Garrett,

Thank you for your report and your advice. We calculated the characteristic timescales of a number of processes of our study. We attach a detailed calculation at the end of this response and focus here on the main points.

First, we focus on the first issue raised by Reviewer 1, concerning the validity of the parcel model and the calculation of the rain rate from the model output.
As we stated before, this first step of our study is an idealized approach, where not all processes and feedbacks are represented in a realistic way. As droplets in the parcel model framework do not rain out, the rain rate is of course not a realistic rain rate, but a proxy for the volume flux of water generated on the parcel through radiation - microphysics interactions.
The method is thus mostly useful for examining the *onset* of drizzle. One can consider the trajectory approach to be an imperfect but useful model (as documented in Stevens, Feingold et al. 1996, JAS, Feingold, Kreidenweis, Zhang, 1998, JGR, Feingold et al. 1999 (GCCN paper)) for examining the combined effect of droplet growth and thermal radiation with and without the radiative effects in a framework that allows for realistic and quantifiable exposure to strong radiative cooling at cloud edges. In contrast, LES allows for for a more faithful treatment of these processes because of the coupling of interactive components, but at the expense of transparency of the radiative effects on drop growth. *In combination the two modeling approaches allow insights that neither could have produced by themselves.*
Note that the above mentioned publications have spelled out the pros and cons of the approach in some detail and we have been careful here not to overstate what can be learned from am ensemble of trajectories.

Text to this effect has been added on Pg 6, line 32 to Pg 7 line 10 in the new manuscript. The direct changes of the text can be seen in the difference pdf file (attached at the end of this text) on Pg 7 line 8 to 22.

Analysis of timescales supports our argument about the usefulness of the approach. The characteristic timescales for a number of processes involved (diffusional droplet growth, diffusional droplet growth with radiation, evaporation, recirculation, buoyancy driven motion and sedimentation show a clear signal: sedimentation occurs on much longer time scales than all the other processes. The timescale of sedimentation was calculated to be in the order of 1 hour, while all the other processes have timescales of a few seconds to a few minutes. This lends credence to the use of the parcel model. Please see the attached page for the specific input values of our calculations. Because there is some arbitrariness in the selection of values we are not convinced that this should appear in the final text.

Regarding the robustness of our results, the influence of 3D radiation on cloud field dynamics is a clear and strong signal (Klinger et al., 2017) but we make no claims that the 3D effect on the microphysics (i.e., rain formation) is clear and strong. The text conveys the message that the thread between the trajectory ensemble and LES results points to a physical explanation for the small increase in the potential (trajectory ensemble) or surface modeled (LES) rain rate. It is highly unlikely that such an effect could be observed in the real world, especially considering the sensitivity of rain to small changes in meteorological conditions (see our reference to the work of Lonitz et al. in the text).

We have made numerous further changes to the text to refine our message and not overstate our claims. The location of the changes in the new manuscript and in the difference pdf are listed at the end of this text.

As noted, we have not included the timescale analysis in the manuscript, in part because the trajectory ensemble approach has a long legacy, and proven utility. An additional concern is that adding it in will break the flow of the text.

We thank you again for your attentiveness in the review process.

Best wishes,
Carolin Klinger

Major changes in the text (Page and line numbers refer to the new manuscript; if major changes were made, the page and line number for the "difference pdf file" (attached at the end of this text) are given as well):

- Pg 6, line 32 – Pg 7, line 10: major changes in the text, please see manuscript or difference pdf file (Pg 7 line 8 to 22)
- Pg 14, line 11/12: added "amount as represented by the the local volume flux of water, or rain rate."
- Pg 17, line 8 and following: major changes to the text, please see manuscript or difference pdf file (Pg 16, line 19 and following)
- Pg 25, line 9 and following: major changes to the text, please see manuscript or difference pdf file (Pg 26, line 4 and following)

Small changes in the text:

- Abstract, Pg 1, line 3  and following: "... is used to describe the onset of rain. The growth of cloud droplets to raindrops is simulated with bin-resolved microphysics... "
- Pg 1, line 5,6:  "raindrop" instead of "rain"
- Pg 1 , line 10: "rain amount" instead of "rain rate"
- Pg 1, line 21: "its" instead of "the droplet's"
- Pg 2, line 17: "studied" instead of "took the next step and looked at"
- Pg 2, line 24: "with a period" instead of "in periods"
- Pg 2, line 30/31: "large eddy simulation (LES)" instead of "LES"
- Pg 4, line 2-5: added "for moist air" and "We note that the radiative cooling is an increasing function of droplet mass."
- Pg 5, line 3: "to derive" instead of "how… is derived"
- Pg 5, line 15: deleted "into each other"
- Pg 5, line 25: "among" instead of "amongst"
- Pg 6, line 7: deleted "The disparity between LES and parcel model microphysics schemes is of no consequence because the parcel model simulations are non interactive."
- Pg 6, line 11: "run" instead of "ran"
- Pg 6, line 27: added "assuming a"
- Pg 8, line 5: "representative" instead of "steady"
- Pg 8, line 10: "on" instead of "in"
- Pg 8, line 14:  "Inevitably" instead of "eventually"
- Pg 8, line 15 "in which" instead of "in that"
- Pg 8, line 16: "spend" instead of "stay"
- Pg 8, line 17: deleted "for BOMEX"
- Pg 8, line 18: "estimated" instead of "estimate"; deleted "the"
- Pg 8, line 21: "cloud" instead of "cooling"
- Pg 10, line 10: "panels" instead of "figures"
- Pg 10, line 14:"the" instead of "an"
- Pg 10, line 16: "in" instead of "at"
- Pg 10, line 17: deleted "in the following"
- Pg 10, line 23: "shown in Fig. 4" instead of "as shown in (Fig. 4)"
- Pg 10, line 31: added "size"
- Pg 11, line 3: "entire" instead of "whole"
- Pg 11, line 9: "recall that the" instead of "the"
- Pg 12, line 4: "wavelengths" instead of "wavelength"
- Pg 12, line 7: "driving force of condensation" instead of "effect on the mass"
- Pg 12, line 11/12: "after which" instead of "before"

- Pg 14, line 5: "prior" instead of "former"
- Pg 16, caption of Fig 11: "occurrences" instead of "occurrence"
- Pg 17, caption of Fig. 12: "Total parcel" instead of "Vertically integrated" and "accounting for drop radii > 20 µm" instead of "by integrating the size bins from 20 µm on"
- Pg 18, line 1: "parcels" instead of "parcel"
- Pg 18, line 2: "recirculation" instead of "recirulation"
- Pg 18, line 2 and 3: "parcel rain rate" instead of "accumulated rain rate"
- Pg 18, line 9: "total parcel" instead of "accumulated"
- Pg 19, caption of Fig. 14: "total rain rate" instead of "vertically integrated"; added " calculated from the trajectory ensemble simulation"; added "cumulated sum"
- Pg 19, line 8: "among" instead of "amongst"
- Pg 19, line 12: "clear" instead of "observed"
- Pg 20, line 2: "noticeable" instead of "considerable"
- Pg 20, line 3: "prior to" instead of "before"
- Pg 23, line 4: deleted "set the"
- Pg 26, line 1: "in recirculating parcels" instead of "from recirculating parcels"
- Pg 26, line 2: "above-mentioned" instead of "above mentioned"
- Pg 26, line 6: added "local"
- Pg 26, line 8: "implications" instead of "implication"

**Diffusional Droplet Growth**

$$\frac{dr}{dt} = \frac{1}{\rho C} \cdot \frac{S}{r} \tag{1}$$

therefore $\tau_{growth}$

$$\tau_{growth} = \frac{r^2 \cdot C}{S} \tag{2}$$

with a droplet radius $r$ of 20 $\mu m$, $C = 1e^{10}\ sm^{-2}$ and a supersaturation $S = 0.01$:

$$\tau_{growth} = 6\ min\ 40\ s \tag{3}$$

**Diffusional Droplet Growth including Thermal Radiative Effects**

$$\frac{dr}{dt} = \frac{1}{\rho C}(\frac{S}{r} - F \cdot E_{net}) \tag{4}$$

therefore $\tau_{growth,rad}$ becomes (when assuming that $r$ is constant over time in the radiative term)

$$\tau_{growth,rad} = \frac{r^2 \cdot C}{S - r \cdot F \cdot E_{net}} \tag{5}$$

with $F \approx 1$ and $E_{net} = \sigma\ T^4 \cdot 0.33 \approx -100\ Wm^{-2}$ (on third of the black body radiation occuring in the window region):

$$\tau_{growth,rad} = 5\ min\ 30\ s \tag{6}$$

**Diffusional Droplet Growth Evaporation**

$$\frac{dr}{dt} = \frac{1}{\rho C} \cdot \frac{S}{r} \tag{7}$$

therefore $\tau_{evap}$

$$\tau_{evap} = \frac{r^2 \cdot C}{S} \tag{8}$$

with a droplet radius $r$ of 20 $\mu m$, $C = 1e^{10}\ sm^{-2}$ and a supersaturation of entrained air $1 - RH$ with $RH = 0.5$ it follows that:

$$\tau_{evap} = 8\ s \tag{9}$$

The timescale including radiative effects is nearly unchanged: $7.96\ s$.

**Recirculation**

$$\tau_{recirc} = \frac{L}{v} \tag{10}$$

with $L = 100\ m$ and $v = 1\ ms^{-1}$:

$$\tau_{recirc} = 1\ min\ 40\ s \tag{11}$$

**Sedimentation**

$$\tau_{sed} = \frac{L}{v} \tag{12}$$

with $L = 100m$ and $v(20\ \mu m) = 0.02\ ms^{-1}$:

$$\tau_{sed} = 1\ h\ 23\ min \tag{13}$$

**Buoyancy Driven Motion**

$$\frac{dw}{dt} = -\rho\frac{dp^{'}}{dz} + B \tag{14}$$

therefore:

$$\tau_{buoy} = \frac{w}{-\frac{1}{\rho}\frac{dp^{'}}{dz} + B} \tag{15}$$

with $\rho = 1\,kgm^{-3}$, $g = 10\,ms^{-1}$ and $B = 0.04\ ms^{-2}$ for $1\,K$ temperature perturbation at $288\ K$, $w = 1\,ms^{-1}$, $dz = 100\,m$ and $dp^{'} = 1000\,Pa$:

$$\tau_{buoy} = 0.1\ s \tag{16}$$

[revised manuscript text omitted]

---

## Author Response (AR3)

Dear Prof. Garrett,

We thank you again for your attentiveness in the review process.

We added the timescale analysis to the manuscript. A discussion of the three important timescales for diffusional droplet growth, diffusional droplet growth with radiation and sedimentation can be found on page 7, line 8-14 of the new manuscript.
The calculation of the timescales can be found in Appendix A, page 27 of the new manuscript.
A difference pdf is attached to this letter.

Best wishes,
Carolin Klinger

[revised manuscript text omitted]